# A predictive model of asymmetric morphogenesis from 3D reconstructions of mouse heart looping dynamics

Jean-François Le Garrec[1,2], Jorge N Domínguez[3†], Audrey Desgrange[1,2†], Kenzo D Ivanovitch[4‡], Etienne Raphaël[1,2], J Andrew Bangham[5], Miguel Torres[4], Enrico Coen[6], Timothy J Mohun[7], Sigolène M Meilhac[1,2]*

[1]*Imagine* - Institut Pasteur, Laboratory of Heart Morphogenesis, Paris, France; [2]INSERM UMR1163, Université Paris Descartes, Paris, France; [3]Department of Experimental Biology, University of Jaén, CU Las Lagunillas, Jaén, Spain; [4]Cardiovascular Development Program, Centro Nacional de Investigaciones Cardiovasculares, CNIC, Madrid, Spain; [5]University of East Anglia, Norwich, United Kingdom; [6]John Innes Centre, Norwich Research Park, Norwich, United Kingdom; [7]The Francis Crick Institute, London, United Kingdom

**Abstract** How left-right patterning drives asymmetric morphogenesis is unclear. Here, we have quantified shape changes during mouse heart looping, from 3D reconstructions by HREM. In combination with cell labelling and computer simulations, we propose a novel model of heart looping. Buckling, when the cardiac tube grows between fixed poles, is modulated by the progressive breakdown of the dorsal mesocardium. We have identified sequential left-right asymmetries at the poles, which bias the buckling in opposite directions, thus leading to a helical shape. Our predictive model is useful to explore the parameter space generating shape variations. The role of the dorsal mesocardium was validated in *Shh*[-/-] mutants, which recapitulate heart shape changes expected from a persistent dorsal mesocardium. Our computer and quantitative tools provide novel insight into the mechanism of heart looping and the contribution of different factors, beyond the simple description of looping direction. This is relevant to congenital heart defects.
DOI: https://doi.org/10.7554/eLife.28951.001

*For correspondence:
sigolene.meilhac@pasteur.fr

†These authors contributed equally to this work

Present address: ‡The Francis Crick Institute, London, United Kingdom

**Competing interests:** The authors declare that no competing interests exist.

## Introduction

Bilateral organisms, such as mammals, are characterised by apparently symmetrical left and right sides. However, left-right patterning of the body is essential for the development of the embryo, and in particular for the formation of visceral organs that are asymmetric. In the case of the heart, the primordium is a tube, which undergoes a process of rightward looping (*Patten, 1922*), thereby acquiring a helical shape (*Männer, 2004*; see *Männer, 2009*). Heart looping, which is the first morphological sign of bilateral asymmetry during embryonic development, is required for the correct alignment of cardiac chambers and thus for the establishment of the double blood circulation. Abnormal left-right patterning is associated with human diseases referred to as heterotaxy, with an incidence of about 1/10 000, including defects in the lung, spleen, liver, stomach, intestine and also complex cardiac malformations, which will determine the prognosis of patients (*Lin et al., 2014*; *Guimier et al., 2015*). Experiments in animal models have provided insight into how left-right patterning is established. The left-right organiser, the node (*Nonaka et al., 1998*), is first detectable in the early embryo, at Embryonic day (E)7.5 in the mouse. Left-right patterning involves a leftward fluid flow, generated by cilia, which initiates a signalling cascade centred on the left determinant

**eLife digest** The heart is an organ that pumps blood throughout the body to supply oxygen and to remove carbon dioxide and waste products. Its left and right side are shaped differently to circulate blood through two pathways: to the lungs and to all other organs.

As the heart develops inside the embryo, it transforms from a simple, straight tube into a helix shape similar to the shell of a snail. During this process called looping, the helix coils anti-clockwise, which determines where the left and right side of the heart form. It is thought that over 20% of heart anomalies in children may be caused by abnormal looping.

Much of what is known about heart development is based on studies in chicken and fish. However, despite its medical significance, it was not fully understood how the heart of mammals acquires its helix shape. Now, Le Garrec et al. were able to investigate the looping process more closely by creating 3D images and computer simulations of the developing mouse heart.

First, Le Garrec et al. studied the cells that build the heart and found that left and right cells contribute differently. For example, the number of cells differed between left and right side. The computer simulations then showed that looping is caused by mechanical constraints, which occur because of the way the heart attaches to the body. These mechanical constraints amplify the differences between left and right cells and cause the heart to acquire an oriented helix shape.

The computer model could predict how the heart shape will change depending on the type of mechanical constraint, or if cells will have varying levels of left/right differences. The model could also accurately reproduce the shape changes observed in the mouse embryo and predict the abnormal shape of embryos with a genetic defect.

The tools generated in this study will help to understand how anomalies could appear as the heart develops in the embryo, and may in the future also be applied to other organs like the gut. A next step will be to explore how genes control the looping of the heart and contribute to heart anomalies in children.

DOI: https://doi.org/10.7554/eLife.28951.002

Nodal, a secretory protein of the transforming growth factor-beta (TGFβ) superfamily (*Levin et al., 1995*; *Collignon et al., 1996*; see *Shiratori and Hamada, 2014*). This initial left-right biasing mechanism is important to coordinate the morphogenesis of visceral organs according to the same reference. However, it does not explain why morphogenesis is asymmetrical. For example, in *Nodal* mutants (*Brennan et al., 2002*), the direction of heart looping is randomised, but the process of looping still takes place. *Brown and Wolpert (1990)* had hypothesised the existence of an additional mechanism, organ specific, for randomly generating asymmetry. Yet the basis of such a mechanism for the looping of the heart tube has remained enigmatic.

Formation of the cardiac tube occurs under the head folds, by fusion of two bilateral heart fields (*Rawles and Rawles, 1943*; *Kinder et al., 1999*). Heart precursors are detected in two waves, referred to as the first and second heart fields (see *Meilhac et al., 2014*). Formation of the cardiac tube, at E8 in the mouse, is initiated on the ventral side such that the heart remains initially attached to the body dorsally, by a tissue referred to as the dorsal mesocardium, which finally breaks down at E9.5. Elongation of the heart tube progresses cranially and caudally by ingression of heart precursors (*Stalsberg, 1969*; *de la Cruz et al., 1977*) and also by proliferation of cells inside the tube (*de Boer et al., 2012*). Growth of the myocardium inside the tube is coherent (*Meilhac, 2003*), suggesting that mixing between cells derived from the right and left heart fields is limited. In agreement with this, cell-labelling experiments have shown that left and right heart precursors contribute to specific regions of the mouse heart tube (*Domínguez et al., 2012*) and later, in the fetal heart, to specific veins, atrial regions or specific great arteries (*Lescroart et al., 2010*; *Lescroart et al., 2012*). Looping of the tube is concomitant with elongation (*Biben and Harvey, 1997*), raising the possibility that the two processes may be linked. Although the general sequence of heart looping has been well studied, the morphological changes of the cardiac tube, particularly in the mammalian embryo, have not been quantified, hindering precise reconstruction of the looping process, as well as description of mutant heart geometries.

In principle, looping of a tube may result from intrinsic (*Noël et al., 2013*; *Taniguchi et al., 2011*) or extrinsic factors (*Davis et al., 2008*; *González-Morales et al., 2015*). In the context of the heart, looping mechanisms have been mainly studied in the chick. Early embryologists made the morphological observation that the elongation of the heart tube is constrained (*His, 1868*). Measurements indicated that the increase of the tube length is much greater than that of the distance between the attached cranial (arterial) and caudal (venous) ends (*Patten, 1922*), leading to the proposal that looping results from a buckling mechanism, when the tube elongates between fixed poles. This mechanism was challenged by explant experiments, showing curvature of isolated heart tubes (*Manning and McLachlan, 1990*), thus pointing to intrinsic rather than extrinsic factors for heart looping. A problem with these analyses lies in the definition of heart looping. Explants generated a C-shape tube, but not a helical shape. In addition, surgical disconnection of a single pole (the arterial pole) impaired heart looping (*Kidokoro et al., 2008*). Thus, the roles of intrinsic and extrinsic factors in heart looping have remained unclear. Theoretically, intrinsic factors may contribute to heart looping, for example by differential growth (*Manasek et al., 1972*) or oriented deformations (*Itasaki et al., 1991*). To predict which 3D shape might emerge from a combination of forces, computer simulations are required. Simulations with a Finite Element Analysis model, in combination with experiments in the chick, have shown how intrinsic and local variations in tissue stiffness or growth can influence phases of heart looping: the increase in cell size ventrally can generate the initial ventral bulge of the heart tube (*Soufan et al., 2006*; *Shi et al., 2014a*), whereas increased growth in the left atrial region can bias the direction of heart looping and generate a rightward C-shape (*Kidokoro et al., 2008*; *Shi et al., 2014b*; *Voronov et al., 2004*). Whether intrinsic factors are dependent on the Nodal left-right signalling cascade remains undetermined, since *Nodal* is transiently expressed in heart precursors and turned off within the heart tube (*Vincent et al., 2004*), and since the Nodal target *Pitx2c*, which is expressed in the heart tube, is not required for heart looping (*Lu et al., 1999*). In addition, the concomitant inflation of cardiac chambers, which is associated with differential growth rates (*de Boer et al., 2012*), as well as oriented growth (*Le Garrec et al., 2013*; *Meilhac et al., 2004*), would interfere with intrinsic factors of heart looping. Therefore, there is currently no convincing demonstration that intrinsic factors are required to produce a helical heart tube shape.

Rather than computer simulations, mechanical simulations with non-biological material such as rubber tubes have explored the theoretical role of extrinsic factors in a buckling mechanism of heart looping, such as a rotation of the tube poles (*Männer, 2004*), or tube growth within the confined space of the pericardial cavity (*Bayraktar and Männer, 2014*). However, removal of the pericardial membrane does not impair heart looping in the longer term (*Kidokoro et al., 2008*; *Nerurkar et al., 2006*), questioning the role of this confinement. The role in heart looping of another mechanical constraint, the attachment by the dorsal mesocardium, was not taken into account in these simulations. Yet, failure of the dorsal mesocardium to break down has been associated with incomplete heart looping, for example in *Shh*[-/-] mouse mutants (*Hildreth et al., 2009*) or upon inhibition of matrix metalloproteinases in the chick (*Linask et al., 2005*). A structure analogous to the dorsal mesocardium of the heart tube, the dorsal mesentery, has been shown to be involved in the looping of another tube, the gut, as a result of left-right asymmetries in its cellular architecture (*Davis et al., 2008*). Left-right asymmetries of the dorsal mesocardium have also been reported (*Linask et al., 2003*; *Linask et al., 2005*), further pointing to this structure as a potential factor of heart looping.

Here, we explore further the buckling mechanism and test whether it is sufficient for heart looping. Using the mouse embryo, we have developed a novel procedure to generate qualitative and quantitative datasets of shape changes in 3D of the looping heart, captured by high-resolution episcopic microscopy (HREM). We provide a novel dynamic 3D computer simulation of heart looping which is based on these data. Our combined modelling and experimental approach shows that asymmetries at the fixed heart poles, generating opposite deformations, associated with the progressive release of the heart tube dorsally, are sufficient to generate looping of a tube growing between fixed poles. Our predictive model is validated in four experimental conditions, using cell labelling, time-lapse imaging and molecular deficiencies. The novel model of heart looping that we propose functions as a generator of asymmetric organ morphogenesis, in the sense of *Brown and Wolpert(1990)*, able to amplify initial left-right differences between heart precursors.

## Results

### 3D reconstruction of the sequence of heart looping in the mouse embryo

We investigated the process of heart looping in the mouse embryo, whereas it had been previously mainly studied in the chick. In litters dissected at E8.5, we observed a spectrum of heart shapes, which suggests that heart looping is a rapid process. Theiler stages, which report general embryonic landmarks over the whole gestation, fail to account for the rapid progression of heart looping. We grouped the embryos according to heart shapes (*Figure 1A*). To order them, we used two criteria. The heart tube is known to elongate by ingression of heart precursors at both the cranial, arterial, pole and the caudal, venous, pole (*Domínguez et al., 2012*; *Zaffran et al., 2004*). Thus, the addition of novel heart regions was taken as an indication of later stages. The cranial addition of the right ventricle and the extension of the outflow tract were particularly striking. Heart looping is also known to correspond to a repositioning of the right ventricle, from an initial cranial to a final right position (*de la Cruz, 1998*). Thus, the position of the right ventricle relative to the left ventricle was taken as another sign of the progression of heart looping. The resulting sequence parallels the addition of somites. However, we found, as others previously (*Kaufman and Navaratnam, 1981*), that heart development was not strictly synchronous with somitogenesis. We propose a novel staging system of early heart development, based on the shape of the heart tube. At E8.5c, the cardiac crescent is visible but the bilateral heart fields have not fused. At E8.5d, the right and left heart fields, adjacent at the midline, bulge separately within the cardiac crescent. At E8.5e, the bilateral heart fields have fused, with a visible midline furrow. They form a primitive left ventricle, which bulges ventrally, with the venous and arterial poles caudal and cranial respectively. This corresponds to the initial cardiac tube, taking the shape of an inverted Y, as the venous pole remains bilateral. At E8.5f, a right ventricular region appears cranially and the tube still appears bilaterally symmetrical. At E8.5g, the first external sign of left-right asymmetry occurs, with the tilting of the tube axis. At E8.5h, the outflow region starts to extend, resulting in a more variable curved heart shape. At E8.5i, the looped tube is clearly detectable; however, the right ventricle has not yet reached its right position. The right ventricle-left ventricle axis does not parallel the embryonic right-left axis. At E8.5j, the right ventricle has reached its final position such that the right ventricle-left ventricle axis tends to be perpendicular to the cranio-caudal axis.

To further characterise the shape changes during heart looping, we acquired 3D images (*Video 1*) by HREM (*Weninger et al., 2006*) and, after image segmentation, reconstructed the 3D shape of the myocardium from E8.5e to E8.5j (*Figure 1B*, and *Figure 1—source data 1*). These 3D reconstructions are essential to extract and quantify geometrical parameters. By plotting the centroid of myocardial sections (*Figure 2A*), we extracted the axis of the cardiac tube (*Figure 2B*). We quantified the increase in the length of the cardiac tube from 183 ± 42 μm (n = 3) at E8.5e to 800 ± 56 μm (n = 3) at E8.5j (*Figure 2C*). We also quantified the repositioning of the right ventricle, by measuring the right ventricle-left ventricle axis relative to the cranio-caudal axis (*Figure 2D*), from 7°±7 (n = 3) at E8.5f to 72°±1 (n = 3) at E8.5j (*Figure 2E*). These measures validate the ordering of stages and provide quantitative references for the evaluation of the progression of heart looping.

### Parameters and simulation of a simple buckling mechanism of heart looping

We analysed the existence of mechanical constraints that have been proposed to influence heart looping, such as the distance between the heart poles. We found that, while the length of the cardiac tube increases 4.4-fold in average between E8.5e and E8.5j, the distance between the arterial and venous poles is constant (p=0.26, Student test between E8.5e and E8.5j) with a value of 146 ± 29 μm in average (*Figure 2C*). These measures are consistent with the buckling mechanism initially proposed by *Patten (1922)*.

To evaluate the shapes that can be generated by a buckling mechanism, we designed a finite element computer model, based on the GPT framework previously used to model flower shapes (*Green et al., 2010*; *Kennaway et al., 2011*). The model, in which the growing tissue is simulated as a continuous sheet of material, can capture tissue deformations at a large scale with a reduced number of arbitrary parameters, compared to cell-based simulations (*Osborne et al., 2017*). Tissue

**A**

| Stage | E8.5c | E8.5d | E8.5e | E8.5f | E8.5g | E8.5h | E8.5i | E8.5j |
|---|---|---|---|---|---|---|---|---|
| Theiler | 12 | 12 | 12 | 12 | 13 | 13 | 13 | 13 |
| Somite nb | 2-3 | 3-5 | 5-6 | 6-8 | 7-8 | 8-10 | 9-11 | ≥ 11 |

**Figure 1.** Stages depicting the progression of heart looping in the mouse. (**A**) Schematic representation of shape changes during the formation and looping of the heart tube (orange) in the E8.5 mouse embryo. Until E8.5f, the mouse embryo appears bilaterally symmetrical, and the heart tube is straight. The staging scale goes from E8.5c to E8.5j (previous E8.5a-b stages are not shown). This scale, focused on the heart, is finer than Theiler stages, and not fully synchronous with the addition of somites. Somite numbers (nb) were counted in a collection of 40 embryos imaged by HREM between E8.5e and E8.5j and of 48 embryos between E8.5c and E8.5d observed under the microscope. (**B**). 3D reconstructions of heart shapes from HREM images at each stage of heart looping. All the reconstructions are aligned with the notochord vertical (green), the arterial and venous poles up and down, respectively. L, left; LA, left atrium; LV, left ventricle; OFT, outflow tract; R, right; RA, right atrium; RV, right ventricle. Scale bar: 100 μm.

DOI: https://doi.org/10.7554/eLife.28951.003

*Figure 1 continued on next page*

*Figure 1 continued*

The following source data is available for figure 1:

**Source data 1.** 3D reconstructions of heart stages during the looping process, in a 3D pdf format.

DOI: https://doi.org/10.7554/eLife.28951.004

deformation depends on the regional growth patterns, modulated in rate and orientation, which are the input parameters of the model. It also depends on the mechanical constraint of maintaining a continuous sheet of tissue elements. In the absence of any asymmetry, the simulation of a tube growing between fixed poles did not lead to buckling, but to a wider tube (*Figure 2F*). When we introduced a small left-right bias (burst of 5% increase in growth at one pole), which would correspond to stochastic, naturally occurring, left-right variations, the tube was able to curve (*Figure 2G*). The buckling mechanism is thus able to generate asymmetric morphogenesis by amplifying small left-right variations. However, in this minimal hypothesis, the tube acquired a C-shape. Therefore, a buckling mechanism is insufficient to account for the biological helical shape of the looped heart tube. This prompted us to further analyse mouse hearts, to quantify left-right asymmetries, as well as additional mechanical constraints.

## Dynamics of the mechanical constraint from the dorsal mesocardium

The heart tube is initially attached dorsally to the body via the dorsal mesocardium. From a mechanical point of view, this may hinder heart looping. We analysed the dynamics of the dorsal mesocardium in our 3D reconstructions. We observed it in serial embryonic sections acquired by HREM and found variations in its lateral thickness along the cranio-caudal axis and between stages (*Figure 3A–C*). Thus, we quantified the thickness of the dorsal mesocardium from the arterial to the venous pole at different stages (*Figure 3D*). The dorsal mesocardium was at least 94 ± 50 μm (n = 3) thick at E8.5e and got thinner to a minimum of 28 ± 13 μm (n = 3 per stage) between E8.5f and E8.5h, corresponding to about two cells. Breakdown of the dorsal mesocardium was detectable from E8.5i, in the arterial half of the tube and progressed towards the venous pole at E8.5j (see thick lines below *Figure 3D*). Before breaking down, the dorsal mesocardium appeared elongated on the dorsal/ventral axis, at E8.5g and E8.5h (*Figure 3E–F*), suggesting stretching of the tissue. The distance to the foregut was higher in the arterial half of the tube, where breakdown of the dorsal mesocardium was initiated. In contrast, the dorsal mesocardium displayed homogenous characteristics in the venous half between E8.5f and E8.5i. Thus, a clear boundary was apparent between the arterial and venous halves of the tube (position 90, *Figure 3D,F*). Our observations show that breakdown of the dorsal mesocardium and heart looping are simultaneous, and we have quantified for the first time the associated spatio-temporal sequence.

## Initial left-right asymmetries during mouse heart looping

Heart looping is a directional event, which depends on left-right patterning. Thus, we examined early left-right asymmetries during the formation of the heart tube. The E8.5f stage, corresponding to the straight heart tube, had always been considered as a stage of bilateral symmetry, as seen externally (*Figure 1*), whereas the chick heart tube undergoes a rightward rotation between stages HH10 and 12 (*de la Cruz et al., 1977*; *de la Cruz, 1998*). We found on HREM sections, that the mouse heart tube at E8.5f was not bilaterally symmetrical, but rather

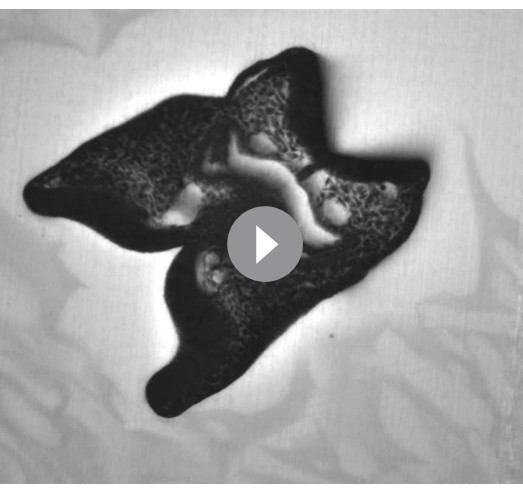

**Video 1.** Example of a 3D stack of HREM images acquired from an embryo at E8.5h. Video showing the successive sections of an E8.5h embryo, at the level of the heart. Images were acquired by HREM every 2 μm.
DOI: https://doi.org/10.7554/eLife.28951.006

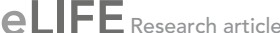

**Figure 2.** The heart tube elongates and loops between fixed poles. (**A**) HREM image of an embryo section at E8.5f, with the notochord (green dot) and the centroid (red dot) of the myocardial tube (pale red) outlined. (**B**) Ventral view of a 3D reconstruction of the heart tube at E8.5i, aligned with the notochord vertical (green), showing the axis of the myocardial tube (red) used for the measurement of its length, and the distance between its poles (blue double-arrow). The measures were taken between the top of the arterial pole and the bifurcation between the two atrial regions at the venous

*Figure 2 continued on next page*

*Figure 2 continued*

pole. The distance between the poles was measured after projection onto the notochord. (C) Comparison between the tube length (red) and the distance between the poles (blue) during heart development. (D) Ventral view of a 3D reconstruction of the heart tube at E8.5h, aligned with the notochord vertical (green), showing the measurement of the orientation of the right ventricle (RV)-left ventricle (LV) axis. The perpendicular (red arrow) to the section of the interventricular sulcus (dotted circle) was drawn and its angle with the notochord (green arrow) was calculated. The measure was taken after projection onto the frontal plane. (E) Inclination of the right ventricle (RV)-left ventricle (LV) axis when looping progresses from a cranio-caudal (0°) towards a left-right orientation (90°). (F) Computer simulation of shape changes, in 3D, with a Finite Element model, seen from ventral (left) and cranial (right) views at step 100. Starting from a straight tube (t = 0), the simulation was run under the hypothesis of a tube growing homogenously between fixed poles, in the absence of any asymmetry. (G) Similar computer simulation, but with a small (5%) left-right asymmetry at one pole, mimicking stochastic, naturally occurring, left-right variations. The coloured scale of longitudinal growth used in the simulations is indicated on the right. Means and standard deviations are shown, with n = 3 for each point. Scale bar: 50 μm (A,B), 100 μm (D).

DOI: https://doi.org/10.7554/eLife.28951.005

rotated towards the right side (*Figure 4A–B*). We quantified this asymmetry, by measuring the right and left angles between the heart tube and the dorsal pericardial wall (*Figure 4C*). We found that this rotation was not present at the previous stage E8.5e. At E8.5f, this asymmetry was specific to the arterial pole (Student test, p<0.001) and not significant in the venous half of the tube (p>0.40). We estimated, at E8.5f in the mouse, that the rotation is in the order of 25° (inset *Figure 4C*). To further follow the rotation of the heart tube, we mapped the position of the limits of the dorsal mesocardium (*Figure 4D–F*). From a ventral view, the ventral aspect of the dorsal mesocardium was shifted to the right compared to the dorsal aspect. This was visible at E8.5f, and amplified at E8.5g, in agreement with the dorso-ventral elongation of the dorsal mesocardium. However, this was specific to the arterial pole and barely visible at the venous pole.

We then analysed potential left-right asymmetries at the venous pole. Whereas the arterial pole was displaced to the right of the midline (notochord), the dorsal mesocardium at the venous pole appeared displaced in an opposite direction, to the left of the midline (*Figure 4F*), in agreement with previous reports of a leftward displacement of the venous pole, both in mouse (*Biben and Harvey, 1997*) and fish (*Chen et al., 1997*). This is not due to an underlying asymmetry in the dorsal pericardial wall: the width of the pericardial wall was found symmetrical throughout the looping process, with a maximum left-sided deviation of 37 μm ± 30 at E8.5i, which is not significant (Student paired test, p>0.09) (*Figure 4—figure supplement 1*). We quantified the position of the venous pole of the heart relative to the notochord and found it significantly left-sided from E8.5g, with a minimum 47 μm ± 22 (E8.5g) and maximum 87 μm ± 10 (E8.5i) deviation (*Figure 5A–B*). We analysed further left-right asymmetries, by addressing cell ingression from the posterior second heart field. Symmetrical dye injections were performed with different colours on the right and left sides (*Figure 5C1–D1*), and the relative contributions to the heart tube were observed (*Figure 5C2–D2*). We evaluated whether there are instances when precursors on one side had been recruited to the heart tube and not that on the other side (*Figure 5E*). This was observed transiently, at stage E8.5g, with a significant higher number of cases when right cells only had been recruited to the heart tube. At a cellular level, left-right differences in cell proliferation have been reported in the sinus venosus at the 7-somite stage (*Galli et al., 2008*), but not in precursor cells. To investigate this further, we analysed cell proliferation with a higher spatio-temporal resolution, using 3D images and our staging system (*Figure 5F–G*). A significant increase in mitotic cells was observed in right precursors of the second heart field at E8.5g, but we did not detect any difference between the anterior and posterior domains.

Our dye injections, proliferation assay and measures of the venous pole position are consistent. We show that there is a leftward displacement and a transient asymmetric cell ingression at the venous pole, which takes place at E8.5g, i.e. one stage after the rotation of the arterial pole at E8.5f. The faster ingression and proliferation of cells on the right side of the venous pole would generate a leftward deformation, opposite and subsequent to the rightward rotation of the arterial pole.

## A novel, predictive, computer model of asymmetric heart morphogenesis

To test how the mechanical constraints and left-right asymmetries that we have observed in the mouse can affect the shape of the cardiac tube, we performed computer simulations. The model

**A**

E8.5f (120μm)

**B**

E8.5g (100μm)

**C**

E8.5i (120μm)

**D**

Lateral thickness of the dorsal mesocardium (μm)

- E8.5e
- E8.5f
- E8.5g
- E8.5h
- E8.5i
- E8.5j

broken down dorsal mesocardium :

**E**

dorsal

R ┼ L

ventral

Foregut

Heart tube

**F**

Elongation of the dorsal mesocardium (μm)

Arterial pole          Position (μm)          Venous pole

**Figure 3.** Progression of dorsal mesocardium breakdown during heart looping. (**A**) Transverse section from an HREM image of an embryo at E8.5f showing the attachment of the heart tube to the body via the dorsal mesocardium (lateral thickness between red arrowheads). The position, along the notochord, of the section is indicated in brackets, as the distance to the bifurcation between the two atrial regions. (**B**) At equidistance between the poles at E8.5g, the dorsal mesocardium is thinner (red arrowheads). (**C**) In the arterial half of the tube at E8.5i, the dorsal mesocardium is broken down

*Figure 3 continued on next page*

*Figure 3 continued*

(open arrowhead). (**D**) Measure of the lateral thickness of the dorsal mesocardium, in serial positions along the notochord, from the arterial pole (left) to the venous pole (right), at successive developmental stages (colour coded). Thick lines below indicate positions where at least one sample had dorsal mesocardium breakdown. The boundary between the arterial and venous halves of the tube is shown by a vertical dashed line. (**E**) Schematic representation of a transverse section, showing the dorso-ventral elongation of the dorsal mesocardium (red double arrow), measured as the distance to the ventral fold of the foregut. (**F**) Measure of the elongation of the dorsal mesocardium, in serial positions along the notochord at successive developmental stages (colour coded). A significant elongation of the dorsal mesocardium, compared to the initial heart tube at E8.5f, is indicated by asterisks (*p-value<0.05 and **p-value<0.01, two-tailed Student test). When the dorsal mesocardium has broken down, the elongation value is set to 0. Means and standard deviations are shown (n = 3 for each stage). Scale bars: 50 μm.

DOI: https://doi.org/10.7554/eLife.28951.007

was initiated as a hollow cylinder of 1800 finite elements, represented by pentahedra (*Figure 6* and *Figure 6—figure supplement 1A–D*). The initial stage represented E8.5f but did not take into account the atrial region which is bifid. To account for the mechanical constraints observed in biological samples, the poles of the cylinder were fixed in the cranio-caudal (z) axis. In addition, the dorsal mesocardium was simulated by constraints along two vertical lines dorsally, restricting displacement in the dorso-ventral (y) and cranio-caudal (z) axes. The breakdown of the dorsal mesocardium was simulated by a progressive release of this constraint, starting from the nodes at mid-length of the tube and progressing towards the poles. The orientation of growth was defined relative to the tube, accompanying its deformation, and not relative to embryonic axes. Longitudinal growth parallels the axis of the tube, whereas circumferential growth is perpendicular. Longitudinal growth was implemented with a baseline value accounting for the observed lengthening of the tube (see *Figure 2C*) and with an initial ventral increase to reproduce the ventral bulge of the cardiac tube. Circumferential growth was added to account for the expansion of each ventricle. The simulation ends after 90 steps, corresponding to stage E8.5i.

To this basal framework, we added growth patterns to simulate left-right asymmetries. The clockwise rotation of the arterial pole was simulated with a left-right difference in circumferential growth, in a tube which is constrained dorsally (*Figure 6B*). It was calibrated to simulate a 25° rotation, which decreases along the tube from the pole down to mid-length, in agreement with biological observations (*Figure 4C*). The asymmetric ingression at the venous pole is simulated with a burst of longitudinal growth on the right (*Figure 6C*). The sequence of growth patterns, which are used as inputs of the model and reflect biological observations, is summarised in *Figure 6A*. There are six input parameters of growth, of which two (the basic longitudinal growth and the circumferential growth at the arterial pole) are set to match observed morphological measurements, three (the initial ventral longitudinal growth, the left and right ventricle circumferential growth) are set to match qualitative shape observations, and one (the longitudinal growth asymmetry at the venous pole) is set arbitrarily to obtain at the end of the simulation a helical shape similar to that observed at stage E8.5i.

Simulations ran with this model showed that, in the context of a tube growing between fixed poles, two left-right asymmetries, generating opposite deformations at the poles, together with the progressive breakdown of the dorsal mesocardium, were sufficient to generate a helical shape (*Figure 6D*), and thus a looped heart tube. The sequence of deformations was compatible with the biological sequence (*Figure 6E–H*, *Video 2*).

We tested whether parameters of the model are required for the acquisition of a helical shape. If the constraint of a fixed distance between the poles was released, simulations led to a C-shape heart tube (*Figure 6—figure supplement 1E*), which is reminiscent of that observed in explant experiments (*Manning and McLachlan, 1990*). If the dorsal mesocardium was not taken into account, simulations also produced a C-shape heart tube (*Figure 6—figure supplement 1F*). If there was no left-right asymmetry, the simulated tube remained straight in a frontal plane, with only the ventral bulge (*Figure 7B* abscissa = 0 and ordinate = 0). If there was a single asymmetry at one pole, again the simulated tube acquired a C-shape (*Figure 7B* abscissa = 0 or ordinate = 0). This indicates that all parameters of the model are required together for generating a helical shape.

Thus, we provide a novel predictive model of heart looping, based on the buckling of the tube, when growing between fixed poles. Buckling is modulated by the progressive breakdown of the dorsal mesocardium and biased by a precise combination of left-right asymmetries at the poles, thus generating a helical shape.



**Figure 4.** Rotation of the arterial pole at E8.5f. (**A**) Transverse section from an HREM image of an embryo at E8.5f, in which the angles (right and left) between the heart tube and the dorsal pericardial wall, at the dorsal mesocardium, are shown. The position of the section is indicated in brackets, as a distance to the bifurcation between the two atrial regions. (**B**) Section of the same heart at the arterial pole, showing the asymmetry of the tube relative to the dorsal-ventral axis of the embryo (white dashed line). (**C**) Quantification of the left-right difference between the angles at the dorsal

*Figure 4 continued on next page*

*Figure 4 continued*

mesocardium, taken at successive positions along the axis of the notochord. A significant difference (***p-value<0.001, paired Student test) at the arterial pole at E8.5f is indicated in grey. From the maximum value of the angular difference, we can estimate the angle of rotation (ρ) of the tube relative to the dorsal-ventral axis (see inset). Means and standard deviations are shown (n = 3 for each stage). (D–E) Transverse embryo sections at E8.5f, at the arterial pole (D) and venous pole (E), with the notochord (green) and the dorsal (red) and ventral (blue) aspects of the dorsal mesocardium outlined. (F) In representative embryos, ventral views, aligned with the notochord, of the position of the dorsal mesocardium along the arterial (top) - venous (down) axis of the tube. The position of the sections in (D) and (E) is indicated. Scale bars: 50 µm (A, B, D, E), 20 µm (F).
DOI: https://doi.org/10.7554/eLife.28951.008
The following figure supplement is available for figure 4:

**Figure supplement 1.** Asymmetry of the dorsal pericardial wall.
DOI: https://doi.org/10.7554/eLife.28951.009

## Exploring the parameter space of the computer model, when left-right asymmetries vary

We explored how variations in initial left-right asymmetries affect the tube shape. The burst of growth at the venous pole may vary in its position or intensity (*Figure 7A*). Computer simulations show that there is a narrow window, in which a helical tube shape can be generated. This corresponds to a burst of longitudinal growth located on the right side, that is generating an opposite deformation to the arterial pole, and a burst intensity defined by the ratio of right over left growth around 2.8-fold. Outside this window, the tube has a flat S shape (in case of a burst on the same position, but with a higher fold difference), or most frequently a C shape. Reversal of the rotation at the arterial pole (leftward), combined with a reversed position of the burst at the venous pole, creates mirror images. We also tested how the intensities of asymmetries at the arterial and venous poles are combined to shape the heart tube (*Figure 7B*). Computer simulations indicate that the intensities of asymmetries at the arterial and venous poles have to be proportional to generate a helical shape. The helix becomes flatter, that is S-shape, when the intensity of asymmetry increases (red dotted diagonal). Such flat S-shapes are also generated close to the window of parameter space corresponding to the helix, whereas C-shapes are observed otherwise. These analyses show that the regulation of the position and intensity of left-right asymmetries at the poles of the heart can greatly influence the shape of the looped heart tube.

## Experimental validation of the rotation of the arterial pole

Computer simulations indicate that acquisition of a helical heart tube requires opposite deformations from the arterial and venous poles. However, mechanistically, computer simulations of heart looping are compatible with either a rightward rotation of the arterial pole, or a burst of growth on the left side, similar (but opposite) to that observed at the venous pole. The computer model can be used to raise predictions and distinguish these cases. In case of a burst of growth at the arterial pole, the model predicts that the initial ventral line of the tube would remain ventral at the end of the simulations (*Figure 8A*), whereas it would shift to the right in case of a rotation (*Figure 8B*). To test these predictions experimentally, we injected DiI at the most cranial ventral midline, at E8.5e (*Figure 8C–D*) and E8.5f (*Figure 8E–F*), that is, before or during the stage when we first observed left-right asymmetry at the arterial pole. After 24 hr of culture, in the looped heart, the label was clearly located on the right side. In addition, we performed time-lapse imaging of mouse embryos at the beginning of heart looping, using inducible reporter lines to better track cell movement in a mosaic tissue. We generated two independent movies, in which a cell could be tracked at the arterial pole between E8.5e/f and E8.5g. In comparison with cells of the headfolds which mainly move caudally, cells of the arterial pole were found to move as much (*Figure 8—figure supplement 1A–C*, *Video 3*) or more (*Figure 8—figure supplement 1D–E*) to the lateral right side. These observations reinforce our morphological measurements (*Figure 4A–C*) and together demonstrate rightward rotation of the arterial pole in the mouse.

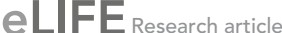

**Figure 5.** Asymmetric cell ingression at the venous pole at E8.5g. (**A**) 3D reconstruction of the heart tube at E8.5i, aligned with the notochord vertical (green), showing the axis of the myocardial tube (red) and the last tube section before the bifurcation (dotted black line) used for the measurement of

*Figure 5 continued on next page*

*Figure 5 continued*

the position of the venous pole. The distance to the notochord (double black arrow) was measured after projection onto the frontal plane. (**B**) Displacement of the venous pole during the looping process. Positive values indicate a leftward deviation, which we take as significant when the null value (0) lies outside the 95% confidence interval (CI) at E8.5g-h and E8.5i, and outside the 99% confidence interval at E8.5i. Means and standard deviations are shown, with n = 3 at E8.5e-h and 4 at E8.5i-j. (**C1–D1**) Brightfield images (ventral view) of embryos at the indicated stage, showing the symmetrical DiI and DiO labelling in the right (white arrow) and left (yellow arrow) posterior heart field, respectively. The contour of the heart tube is outlined (dotted white line). (**C2–D2**) Fluorescent signal on brightfield images (dorsal view) of the isolated heart after 24 hr of embryo culture, in which the labelled regions deriving from the right (white arrows) and left (yellow arrows) injections are visible. The boundary of the heart tube is indicated (dashed white line), showing that both labelled regions are within the heart tube in C2, whereas this is the case only for the right label in D2. (**E**) Quantification of left-right differences in the ingression of cells from the posterior heart field. At each stage of labelling, the number of embryos, in which the right (white) or left (yellow) label had progressed further into the heart tube after 24 hr of culture, is indicated. Embryos in which no obvious asymmetry between the right and left was observed were equally distributed in the two other categories. A significant difference is observed at E8.5g only (chi-square test with Yates' correction, n = 11(E8.5d), 10(E8.5e), 17(E8.5f), 24(E8.5g), 9(E8.5h), p-value=1(E8.5d), 0.21(E8.5e), 0.81(E8.5f), <0.01(**, E8.5g), 1(E8.5h). Raw counts are presented in the table below. dRA: dorsal right atrium; dLA: dorsal left atrium; OFT: outflow tract. (**F**) Whole mount immunostaining of an embryo at E8.5g, showing the second heart field domains, marked by Isl1 (green), used for the quantification of mitotic cells (red), labelled with phosphorylated histone H3 (P–H3). The limit between the anterior (AHF) and posterior (PHF) heart fields is taken as the middle between the heart poles (see *Figures 3D–E* and *4C*). The midline is highlighted by a vertical dotted line. The left headfold (hf) was cut and used as a landmark to orient samples. Examples of optic tissue slices (1, 2) are shown on the right, at the level indicated by the horizontal lines, with arrows (white, right cells, yellow, left cells) pointing to mitotic second heart field cells. Asterisks indicate unspecific bright staining (red asterisk), in yolk sac edges and the foregut cavity, which was removed in the left panel (white asterisk) for better visualisation of the second heart field. (**G**) Quantification of the number of mitotic cells in the second heart field, shown for individual embryos (n = 2, upper and lower lanes), indicating a significant increase on the right side (chi-square test, p-value=0.02 for the upper embryo and p=0.002 for the lower embryo), with no difference between the anterior and posterior domains. Scale bars: 100 µm.

DOI: https://doi.org/10.7554/eLife.28951.010

## Simulation and experimental validation of the role of the dorsal mesocardium

Our morphological observations showed that breakdown of the dorsal mesocardium and heart looping are simultaneous. To further explore how the dorsal mesocardium influences heart shape, we used the computer model to raise predictions. If the dorsal mesocardium is persistent, computer simulations indicate that looping of the heart tube is severely affected (*Figure 9A*, *Video 4*). The right ventricle fails to reach a right position relative to the left ventricle and the curvature of the tube is abnormal. This is reminiscent of *Shh*$^{-/-}$ mutants, in which heart looping was reported to be incomplete and the dorsal mesocardium persistent (*Hildreth et al., 2009*). Thus, to validate experimentally the prediction of our model, we analysed in more detail the heart tube of *Shh*$^{-/-}$ mutants, and compared them to our simulations. Qualitatively, the shape observed in *Shh*$^{-/-}$ mutants was similar to the shape predicted by the computer model (*Figure 9A–B*). Reconstruction of the axis of the heart tube from HREM images showed that *Shh*$^{-/-}$ mutant hearts fail to acquire a helical shape, at a stage when control hearts are looped (*Figure 9C* and *Figure 9—source data 1*). This is associated with a persistent dorsal mesocardium (*Figure 9D–E* and *Figure 9—figure supplement 1*). Between 10 and 12 somite stages, whereas all control embryos displayed a broken down dorsal mesocardium, this concerned only 2/5 mutants (*Figure 9—figure supplement 1A,C*). This was not a delay, as at E9.5, 2/6 mutants had no sign of dorsal mesocardium breakdown, and in the other 4/6 mutants, dorsal mesocardium breakdown was detected over a maximum length of 30 µm, which is much less than a breakdown over a minimum of 124 µm in control littermates (*Figure 9—figure supplement 2*). At E8.5, the dorsal mesocardium of mutant embryos was elongated in the dorso-ventral axis, specifically in the arterial half of the tube, however, with higher values in mutant compared to control samples (*Figure 9—figure supplement 1C,D*). These data indicate that the dorsal mesocardium is regionalised properly in *Shh*$^{-/-}$ mutants, and potentially able to stretch, but fails to largely break down.

As the tube of *Shh*$^{-/-}$ mutants is not straight, it suggests that some looping has occurred. The direction of the loop was found normally rightward (n = 5/5 at E8.5, 6/6 at E9.5) (*Figure 9—source data 1* and *Figure 9—figure supplement 2*). However, heart looping was severely impaired in *Shh*$^{-/-}$ mutants (*Figure 9F*). When we measured the tube length, we found that *Shh*$^{-/-}$ mutants have a significantly shorter heart tube, compared to control samples (*Figure 9G*). Lengthening of the tube is not arrested, as the increase in the length of the heart tube relative to the number of somites follows a linear regression, with a slope not significantly different from that of control samples. This suggests

**Figure 6.** Computer simulations of heart looping integrating mechanical constraints and left-right asymmetries. (**A**) Timeline of the simulations, with the successive events reflecting experimental observations. (**B**) Simulation of heart shape changes in 3D with a Finite Element model of a straight tube, seen from left (top) and right (down) views. The arterial pole asymmetry is modelled as a circumferential contraction (negative circumferential growth) on the right side and an expansion (positive circumferential growth) on the left side, both in a gradient towards the mid-length of the tube. The asymmetry at the arterial pole is calibrated to generate a 25° rotation. Simultaneously, the inflation of the ventricular chambers and the breakdown of the dorsal mesocardium (black bars), from the mid-length of the tube, are initiated. Where the dorsal mesocardium is present, the tube is free to move only along the x direction (f(x)). Where it has broken down, the tube is free to move along all the three directions (f(x,y,z)). (**C**) The venous pole asymmetry is modelled as a difference (2.8-fold) in longitudinal growth between the right and the left side. At this stage, breakdown of the dorsal

*Figure 6 continued on next page*

*Figure 6 continued*

mesocardium has progressed towards the poles, and thus, the tube is free along half of its length. (D) 3D shape of the cardiac tube at the end of the simulation showing the typical helix of the looped heart tube. The right (RV) and left (LV) ventricles are in darker and lighter blue, respectively. (E–H) Comparison of biological and simulated shapes. (E) Ventral views of 3D reconstructions of the heart tube at E8.5g, when looping begins. On the right, the myocardial layer (yellow) is made transparent, revealing the tube axis (red), and the notochord behind (green). (F) Ventral view of a simulated heart tube at step 60. The axis of the tube is shown as a red line. (G) Ventral views of 3D reconstructions of the heart tube at E8.5i, when looping has progressed to a counter-clockwise helix (as seen from the arterial pole). (H) Ventral view of a simulated heart tube at step 90.

DOI: https://doi.org/10.7554/eLife.28951.011

The following figure supplement is available for figure 6:

**Figure supplement 1.** Parameters of the computer model.

DOI: https://doi.org/10.7554/eLife.28951.012

that defects in heart looping in *Shh*[-/-] mutants are not the result of overall reduced growth. In contrast, our computer simulations show that persistence of the dorsal mesocardium constrains the longitudinal growth of the heart tube, such that enlargement of the tube is observed to account for the same amount of overall tissue growth. Thus, we measured the perimeter of the heart tube in *Shh*[-/-] mutants and found a significant 1.2-fold increase compared to controls (*Figure 9H*, Student test, p=0.03). The overall area of the heart tube did not change significantly between mutants and controls (0.56 ± 0.12 mm$^2$ and 0.55 ± 1.0 mm$^2$, respectively, n = 5 and 5, Student test, p=0.75), further showing that there is no overall growth defects in *Shh*[-/-] mutant hearts. We also confirmed that the distance between the poles is not significantly changed in *Shh*[-/-] mutant hearts (121 ± 15 µm, compared to 132 ± 17 µm in controls, n = 5 and 5, p=0.31, Student test). Finally, computer simulations predict that persistence of the dorsal mesocardium constrains the curvature of the tube axis in the transverse plane. This prediction is validated in *Shh*[-/-] mutant hearts, in which we observed a narrower curvature of the tube axis in the transverse plane (*Figure 9H–J*). Alternatively, we tested whether the shape of *Shh*[-/-] mutant hearts could be explained by impairment of left-right signalling: a reduction by 50% of left-right asymmetries would account for the malposition of the right ventricle, but not for the changes in length and the narrow helix (see transverse sector) of the heart tube (*Figure 9—figure supplement 3*). In summary, there is a discrepancy between the real shape of *Shh*[-/-] mutant hearts and the simulated shape in case of reduced left-right signalling. In contrast, there is a good match between the shape of *Shh*[-/-] mutant hearts and the simulated shape in case of persistent dorsal mesocardium, both qualitatively and quantitatively for the change of values of four geometrical parameters.

Another case of dorsal mesocardium persistence, without manipulating Shh, was analysed. In the chick, failure of dorsal mesocardium breakdown associated with incomplete heart looping was reported upon inhibition of matrix metalloproteinases (Mmp) (*Linask et al., 2005*). We investigated whether this is the case in the mouse. We detected expression of Mmp2 in the foregut of the mouse at the beginning of heart looping (E8.5g), with vesicular localisations in the cardiac region, suggesting secretion of Mmp2 (*Figure 10A*). To test the role of Mmp, we used GM6001, a broad-spectrum Mmp inhibitor (*Shichi et al., 2011*), in cultures of mouse embryos at E8.5d-e (*Figure 10B*). After 10 hr of culture, the embryos reached stage E8.5h-i in the control situation (DMSO) (*Figure 10C*). Whereas all control embryos (3/3) displayed a broken down dorsal mesocardium, this concerned only 2/4 mutants (*Figure 10D–E*). Dorsal mesocardium breakdown was detected over a maximum length of 8 µm (3/4) upon GM6001 treatment, which is

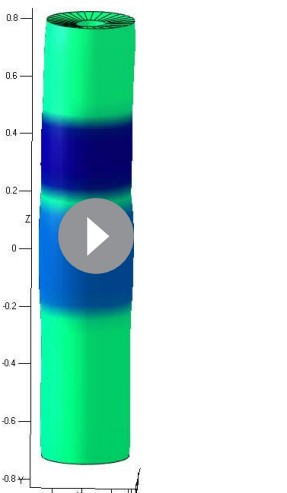

**Video 2.** Computer simulation of heart looping in the control situation.

DOI: https://doi.org/10.7554/eLife.28951.013



**Figure 7.** Variations in left-right asymmetries at the poles impact heart shape. In the computer simulation, parameters of pole asymmetries were explored to analyse the variety of output shapes of the heart tube. Simulated shapes are shown at step 80 with the ventral line in red, from ventral (left) and cranial (right) views. The inflation of ventricular chambers is not simulated here. (A) In this series of simulation, the venous pole asymmetry is explored, with varying positions (abscissa) and varying intensities (ordinate). The venous pole asymmetry is simulated as a burst of growth (red dot). Its

*Figure 7 continued on next page*

*Figure 7 continued*

position is shown relative to the dorsal mesocardium (position 0°, with the left at 90°). The intensity of the burst is defined as a growth increase with the indicated fold difference relative to the opposite side. The angle of rotation of the arterial pole is fixed (25°), but variation in its direction is considered (rightward and leftward in the upper and lower portions of the ordinate respectively). The central green diamond indicates a centre of symmetry of the diagram. (**B**) In this series of simulation, the relationship between asymmetry intensities at the venous (abscissa) and arterial (ordinate) poles is explored. The venous pole asymmetry is simulated as in (**A**), with a fixed position (270°). The arterial pole asymmetry is simulated by a rotation of varying angle. The diagonal, which indicates proportional increase in asymmetries at the arterial and venous poles, is labelled as a red dotted arrow. The biological heart loop, which is a counter-clockwise helix (as seen in a cranial view), corresponds to a narrow window of parameter space, which is highlighted in green.

DOI: https://doi.org/10.7554/eLife.28951.014

much less than a breakdown over 23–125 µm in control cultures (3/3). In this situation again, persistence of the dorsal mesocardium was associated with incomplete heart looping. After 24 hr of culture, the right ventricle failed in most cases (7/9) to reach a right position relative to the left ventricle (*Figure 10F*), leading to a tube shape similar to that predicted by the computer model (*Figure 9A*). Taken together, our analyses combining two experimental conditions and computer modelling highlight the important role of the dorsal mesocardium in heart looping.

## Discussion

We have reconstructed for the first time heart looping in 3D, taking the mouse as a model. We provide novel tools to quantify the progression of heart looping and analyse the 3D shape. In addition, we have characterised the spatio-temporal sequence of mechanical constraints during looping, including the elongation of the tube, the distance between the poles and the breakdown of the dorsal attachment. With additional left-right asymmetries uncovered at the arterial and venous poles, we propose a novel predictive model of heart looping.

Our staging system extends that proposed previously (*Biben and Harvey, 1997*). LS-0 is equivalent to E8.5e/f, when the tube axis parallels the cranio-caudal axis. LS-I is equivalent to E8.5g, when the initial tilting of the tube axis is detected, and the venous pole is displaced leftward. Although we did not use the atrioventricular sulcus as a marker of staging, it can be monitored from our 3D reconstructions (*Figure 1*, *Figure 1—source data 1*), showing that LS-II is equivalent to E8.5g/h. LS-III, which corresponds to ventricular looping, is equivalent to E8.5h-j. We now provide 3D reconstructions and quantitative spatio-temporal measures to assist staging of a continuous process. Several geometrical parameters can be used to assess the progression of heart looping, such as the length of the heart tube, the right ventricle/left ventricle orientation, the rotation of the arterial pole, the position of the venous pole, the extent and localisation of the dorsal mesocardium breakdown. These tools will be instrumental to analyse defects of heart looping beyond the simple description of looping direction, as reported previously in mutant embryos. Our model of heart looping predicts that defects in left-right patterning not only result in impaired direction of heart looping, but also in different shapes, from C-shapes to flat S-shapes. Our tools open novel avenues to study heart looping defects with a greater precision and thus, to better understand the mechanism of looping and the respective contribution of different factors.

Our model, which is based on analyses in the mouse, could potentially be applied to heart looping in other species. In the mouse, we show that the distance between the poles of the tube is fixed, whereas the tube length increases 4.4-fold during the process of looping. We detect a rightward rotation of the arterial pole, which precedes a leftward displacement and deformation of the venous pole. These features are conserved between chick and mouse, despite variations in the morphological sequence of heart looping. The distance between the poles of the chick heart tube has long been reported to be fixed (*Patten, 1922*). However, the value is higher in the chick (600–800 µm) compared to the mouse (100–200 µm), and thus, a straight heart tube is more obvious in the chick. The fold increase in the heart tube length is comparable in the chick and mouse (*Patten, 1922*). A rightward rotation of the ventricular region in the chick was shown by labelling with iron oxide particles (*de la Cruz et al., 1977*; *de la Cruz, 1998*), preceding the leftward displacement of the venous pole (*Kidokoro et al., 2008*). This is compatible with mechanical simulations showing that opposite and sequential rotations of the heart poles are sufficient to generate a helical tube shape

| Ventral view | Right view |
|---|---|

**A**

**B**

**C**

**D**

E8.5e

**E**

E8.5f

**F**

**Figure 8.** Model prediction and experimental validation of the rotation of the ventral line. (A–B) Computer simulations of heart looping with the fate of the initial ventral line shown in blue. The simulated shape is shown at

*Figure 8 continued on next page*

*Figure 8 continued*

step 90 in ventral (left) and right-lateral (right) views. When the asymmetry at the arterial pole is simulated by differential longitudinal growth between the left and right, the blue line remains ventral throughout the simulation (**A**). When the asymmetry at the arterial pole is simulated by a rightward rotation (25°), as in *Figure 6*, the blue line is displaced to the right (**B**). (**C–D**) An example of a heart at E8.5e is shown on the left in C, just after DiI labelling at the most cranial ventral midline (white arrow). This is schematised on the left in **D**. Fluorescent signal on brightfield images of hearts, after 24 hr culture are shown on the right. The initial ventral line has been displaced to the right (n = 2/2). (**E–F**) Similar images of hearts, after 24 hr culture of an E8.5f embryo, labelled by DiI at the most cranial ventral midline (see scheme). The initial ventral line has been displaced to the right (n = 4/5). Scale bars: 100 µm.

DOI: https://doi.org/10.7554/eLife.28951.015

The following figure supplement is available for figure 8:

**Figure supplement 1.** Live-imaging of the rotation of the arterial pole.
DOI: https://doi.org/10.7554/eLife.28951.016

---

(*Männer, 2004*). These simulations postulated a 90° rotation, in agreement with the displacement of iron oxide particles in the outer curvature of the chick heart. However, we have measured a 25° rotation of the arterial pole in the mouse. This could be a species difference, because the C-shape of the chick heart tube observed at stage HH12 (*de la Cruz, 1998*; *Männer, 2000*) has no equivalent in the mouse. Alternatively, it is possible that confinement in the pericardial cavity increases the apparent rotation of the outer curvature of the heart, generating a transient C-shape heart tube in the chick. This is supported by the reduced displacement of the outer curvature upon removal of the pericardial membrane (splanchnopleura) in the chick, with no consequence for heart looping in the longer term (*Voronov and Taber, 2002*; *Nerurkar et al., 2006*). Rotation of the arterial pole in the chick would need to be measured with the same approach as described here, that is the angle between the heart tube and the dorsal pericardial wall. At the venous pole of the mouse, we do not detect a significant leftward rotation, as proposed by *Männer, 2004*, but rather an asymmetric ingression and proliferation of heart precursors, in favour of right cells. This is in keeping with the increased size of the right atrium compared to the left at E8.5 (*Meilhac et al., 2004*). Cell labelling experiments in the posterior second heart field at E8.5 showed that right cells contribute not only to the right atrium, but also to the ventral aspect of the left atrium (*Domínguez et al., 2012*), indicating a wider deployment of right cells in the mouse. An asymmetric ingression of heart precursors, in favour of right cells, has also been observed in the chick by grafting experiments at the beginning of heart looping (HH10, *Stalsberg, 1969*). These similarities suggest that our model would also apply to the chick embryo, so that the mechanism of heart looping may be conserved in amniotes. In the zebrafish, formation of the heart tube follows a different morphogenetic process, involving the formation of a cone, which telescopes out to form a tube (*Stainier et al., 1993*). The looped heart tube in the zebrafish is a flat-S, distinct from the helix in chick and mouse. The fish tube does not grow between fixed poles, has no equivalent of the dorsal mesocardium, and looping was shown to depend on intrinsic factors (*Noël et al., 2013*), thus ruling out a buckling mechanism. However, left-right asymmetry of the zebrafish heart tube is first seen at the venous pole, with a leftward displacement of the atrial region referred to as cardiac jogging (*Chen et al., 1997*), resulting from asymmetric migration and involution of cells in the cardiac cone (*Baker et al., 2008*; *Rohr et al., 2008*; *Smith et al., 2008*). It is about a day later that the ventricle, at the arterial pole, bends rightward, in a process referred to as cardiac looping (*Chen et al., 1997*; *Stainier et al., 1993*). Thus, as in chick and mouse, opposite and sequential deformations occur at the poles of the zebrafish heart tube. However, the order is reversed, with a venous pole displacement first in fish, whereas

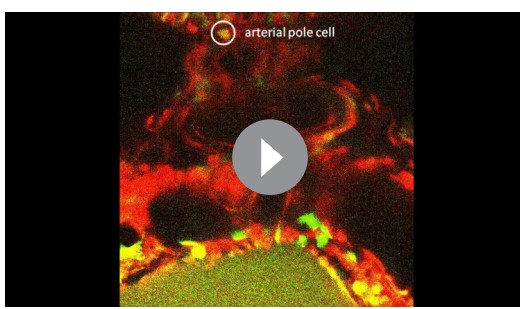

**Video 3.** Example of live-imaging of the arterial pole rotation. Video related to *Figure 8—figure supplement 1A–C*.
DOI: https://doi.org/10.7554/eLife.28951.017



**Figure 9.** Model prediction and experimental validation, in the case of persistent dorsal mesocardium. (**A**) Computer simulations of heart looping in control conditions (left), as in *Figure 6*, and when the dorsal mesocardium is persistent (right, simulated mutant). Simulated shapes are shown at step 90 in a ventral view, with the right (RV) and left (LV) ventricles in darker and lighter blue, respectively. The right ventricle-left ventricle axis (red dotted line) remains closer to the cranio-caudal axis (green dotted line) in the simulated mutant compared to control. (**B**) 3D visualisation of HREM images of a

*Figure 9 continued on next page*

*Figure 9 continued*

control (left) and *Shh*[-/-] mutant (right) embryo at E8.5, seen in a ventral view. (C) 3D reconstructions of the hearts shown in B, aligned with the notochord vertical (green). The axis of the myocardial tube is highlighted in red. (D) HREM section of a control embryo (left, E8.5j, 11 somites) showing the broken down dorsal mesocardium (red open arrowhead), and of a *Shh*[-/-] mutant embryo (right, 12 somites), showing persistence of the dorsal mesocardium (red arrowhead) for a myocardial tube of similar length (549 µm in the control and 535 µm in the mutant). The position of the section along the notochord is indicated in brackets. (E) 3D reconstructions from HREM images of a control (left) and a *Shh*[-/-] mutant (right) heart in ventral (top) and dorsal (bottom) views. The extent of the dorsal mesocardium is shown as a red bar on the side. (F) Graph showing the orientation of the right ventricle-left ventricle axis in control (n = 8) and *Shh*[-/-] mutants (n = 5) at 10 to 12 somite stages. (G) Graph showing the increase in the length of the myocardial tube as a function of the somite number, in control (blue) and *Shh*[-/-] mutant (red) embryos. The least-square linear regression lines and the Pearson coefficient $R^2$ are shown for both the controls (continuous line, n = 22) and mutants (dotted line, n = 5). The two slopes are not significantly different, but the y-intercepts are (ANCOVA, p-value=0.001), indicating that the mutants have a significantly shorter tube length. (H) Validation of quantitative predictions raised by computer simulations for four geometrical parameters. (I) Cranial views of simulated hearts (left, control, right, mutant), showing the tube axis in red and the transverse sector in which it is inscribed in white. (J) 3D reconstructions from HREM images of a control (left) and *Shh*[-/-] mutant (right) heart, shown in a cranial view. The axis of the myocardial tube is in red, and the transverse sector in which it is inscribed is shown as red dotted lines. (K) Quantification of the angles of these transverse sectors in controls (blue, n = 8) and *Shh*[-/-] mutants (red, n = 5). The heart tube helix is significantly wider in control samples (*p-value<0.05, Mann-Whitney U test). Means and standard deviations are shown. Scale bars: 100 µm.

DOI: https://doi.org/10.7554/eLife.28951.018

The following source data and figure supplements are available for figure 9:

**Source data 1.** 3D reconstructions of all hearts of *Shh*[-/-] mutants and control littermates analysed at E8.5.
DOI: https://doi.org/10.7554/eLife.28951.022

**Figure supplement 1.** Structure of the dorsal mesocardium in *Shh*[-/-] mutants and control littermates at E8.5.
DOI: https://doi.org/10.7554/eLife.28951.019

**Figure supplement 2.** *Shh*[-/-] mutants and control littermates at E9.5.
DOI: https://doi.org/10.7554/eLife.28951.020

**Figure supplement 3.** Alternative simulations and comparison with *Shh*[-/-] mutant hearts.
DOI: https://doi.org/10.7554/eLife.28951.021

---

it is the arterial pole deformation first in amniotes. Further analyses of the molecular pathways and cell behaviour underlying asymmetric heart morphogenesis in different animal models will be essential to further assess the degree of conservation of the looping mechanism in vertebrates.

The molecular and cellular mechanism of heart looping in amniotes remains poorly understood. Left-right asymmetric cell proliferation, in favour of right cells and depending on the Nodal target *Pitx2c*, has been observed in the sinus venosus of the mouse at the seven somite stage (*Galli et al., 2008*). We extend the observation of asymmetric proliferation to precursor cells of the second heart field, indicating a potential mechanism for the asymmetric cell ingression that we have observed in the venous pole at E8.5g. The rotation of the arterial pole that we detect in the early heart tube at E8.5f, precedes the well-known rotation of the outflow tract, observed with transgenic markers from E9.5 (*Bajolle et al., 2006*) and the later spiralling of the aorta and pulmonary trunk. It is possible that a continuous process of rightward rotation at the arterial pole takes place throughout heart morphogenesis. However, we can only speculate about its mechanism, by homology with other examples of asymmetric tubular morphogenesis: movements of myocardial cells in the zebrafish result in a rotation of the cardiac cone (*Baker et al., 2008*; *Rohr et al., 2008*; *Smith et al., 2008*), tilting of the gut tube in the mouse depends on asymmetries in the cellular architecture of the dorsal mesentery (*Davis et al., 2008*), and in *Drosophila*, rotation of the hindgut or genitalia is associated with the chiral activity of an atypical myosin, Myo1D, in an

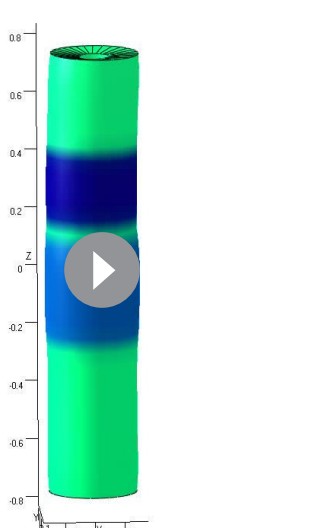

**Video 4.** Computer simulation of heart looping in the case of persistent dorsal mesocardium.
DOI: https://doi.org/10.7554/eLife.28951.023



**Figure 10.** Matrix metalloproteases are required for dorsal mesocardium breakdown and heart looping. (A) Immunostaining of Mmp2, showing expression in the foregut (fg) at E8.5g, and vesicular localisations in the cardiac region (arrowheads in A1), but not in the neural tube (nt, A2). Maximum intensity projections are shown, over 14 (A), 35 (A1) or 31 (A2) z-sections. (B) Scheme of the experimental procedure. (C) Brightfield images of embryos after 10-hr culture with GM6001, an inhibitor of matrix metalloproteases, or the adjuvant (DMSO). (D) Quantification of the lateral thickness of the dorsal mesocardium, measured as in *Figure 3D*, in 3 DMSO (up) and 4 GM6001 (down) treated embryos after 10 hr of culture. Samples are identified with a number (e), and the somite number (So) of the embryo is indicated in brackets. (E) HREM sections of embryos showing a broken down dorsal mesocardium (red open arrowhead) upon DMSO treatment, and a persistent dorsal mesocardium (red arrowhead) upon GM6001 treatment. The position of the section along the notochord is indicated in brackets. (F) Brightfield images of embryos after 24 hr culture. The frequency of normal looping, in the presence of DMSO, or abnormal heart looping, in the presence of GM6001, is indicated in the lower right corner. The contour of the heart tube is outlined. LV, left ventricle; RV, right ventricle. Scale bars: 50 µm (A), 200 µm (C, E–F).
DOI: https://doi.org/10.7554/eLife.28951.024

organiser tissue (*González-Morales et al., 2015*; *Spéder et al., 2006*). The asymmetric (right > left) proliferation of precursor cells was observed also in the anterior domain of the second heart field, suggesting potential modulation of the arterial pole deformation at E8.5g. This will require further analysis. Breakdown of the dorsal mesocardium was shown, in the chick, not to be associated with apoptosis, but rather to depend on the activity of matrix metalloproteinases (*Linask et al., 2005*). This is in agreement with our mouse embryo cultures treated with the inhibitor GM6001 and suggests a mechanism of degradation of the extra-cellular matrix. Our quantification of the breakdown of the dorsal mesocardium, as well as of the tube rotation, uncovers a sharp boundary between the arterial and venous halves of the mouse cardiac tube at E8.5. This is in keeping with the regionalisation of the second heart field, lying in the dorsal pericardial wall, into anterior and posterior domains. Such regionalisation has been characterised later, at E9.5, based on epithelial markers (*Francou et al., 2014*) and differential gene expression, for example of the transcription factors Tbx1 and Tbx5, respectively (*Rana et al., 2014*). Our results highlight an earlier stage of this boundary, which would provide a context for the differential left-right asymmetries that we observe at the poles of the heart, leading to opposite deformations (rightward at the arterial pole and leftward at the venous pole). Thus, heart looping integrates both left-right and anterior-posterior patterning.

Our work identifies Shh as an upstream regulator of the breakdown of the dorsal mesocardium. Hh signalling was shown to be required in the lateral plate mesoderm for the establishment of left-right patterning. However, in this context, Shh and Ihh ligands are redundant, so that it is only in the double mutant of the ligands (*Shh^{flox/-}; Ihh^{flox/-}; Sox2-Cre*) or in the mutant of the receptor (*Smo^{-/-}*) that left markers are absent in the lateral plate mesoderm (*Tsiairis and McMahon, 2009*). Shh is also required for the formation of the floorplate, which functions as a barrier for the maintenance of left-right asymmetry (*Meyers and Martin, 1999*). In *Shh^{-/-}* mutants, *Nodal* expression is initiated correctly, whereas the downstream target *Pitx2* is bilaterally expressed (*Hildreth et al., 2009*; *Meyers and Martin, 1999*). This is in agreement with our observation that the incomplete looping of *Shh^{-/-}* mutants was biased correctly towards the right side, as also previously reported (*Tsukui et al., 1999*), and distinct from cases of bilateral *Nodal* expression leading to a randomised looping direction (*Murray and Gridley, 2006*; *Izraeli et al., 1999*; *Furtado et al., 2008*). From these data, we conclude that left-right patterning is not disrupted initially in *Shh^{-/-}* mutants. Our simulations with reduced left-right signalling fail to reproduce the heart shape observed in *Shh^{-/-}* mutants. Shh has been shown to regulate cell differentiation in the second heart field (*Goddeeris et al., 2008*; *Zhang et al., 2001*), thus affecting the growth of the heart. However, during looping, we have not detected any significant change in the heart growth of *Shh^{-/-}* mutants, indicating that looping defects are not related to defective growth. The good quantitative match between our computer simulations and mutant shapes rather show that heart looping defects in *Shh^{-/-}* mutants recapitulate that expected from a persistent dorsal mesocardium. This is reinforced by the observation of similar looping defects associated with dorsal mesocardium persistence in another experimental condition, upon matrix metalloproteinase inhibition.

Our model of heart looping extends the buckling mechanism, theoretically proposed by early embryologists (*Patten, 1922*) and tested mechanically with non-biological materials (*Männer, 2004*; *Bayraktar and Männer, 2014*). We now integrate another mechanical constraint, the dorsal mesocardium, as well as left-right asymmetries taken from biological observations. The direction of the buckling is biased by left-right asymmetries, whereas the degree of buckling depends on the forces applied at the ends, i.e. the magnitude of growth of the heart tube, and restriction from the dorsal mesocardium. The importance of growth of the cardiac tube, as a pre-requisite for the buckling, is supported experimentally by the looping defects observed in mouse mutants with reduced ingression of heart progenitors, when the cardiomyocyte differentiation cascade is affected (see mutants for *Nkx2-5* [*Lyons et al., 1995*], *Mef2c* [*Lin et al., 1997*], *Isl1* [*Cai et al., 2003*], *Tbx20* [*Stennard et al., 2005*] or *Tbx3* [*Ribeiro et al., 2007*]). The spatio-temporal sequence of breakdown of the dorsal mesocardium, relative to the growth and asymmetries of the heart tube, appears, in our computer model, as an important determinant of embryonic heart shape. This is supported by our analysis of heart looping defects in *Shh^{-/-}* mutants or GM6001-treated embryos, in which the dorsal mesocardium is persistent. Our computer simulations also show that left-right asymmetries extrinsic to the heart tube, that is in heart precursors, are sufficient for heart looping. This suggests that intrinsic asymmetries, such as complex growth gradients or growth orientations within the heart tube may be largely dispensable for heart looping. This is in agreement with current knowledge of

left-right patterning of cardiac cells. Expression of the major left determinant Nodal is detected in heart precursors, and not within the heart tube (*Collignon et al., 1996*; *Vincent et al., 2004*), whereas the Nodal target *Pitx2c*, which is expressed in the heart tube, is not required for heart looping (*Lu et al., 1999*). Our computer simulations show that the position and intensities of left-right asymmetries at the poles of the heart greatly influence heart shape. Thus, our model provides a mechanism for a generator of asymmetric morphogenesis, specific to the heart, able to amplify variations in left-right patterning. The existence of such a mechanism had been postulated by *Brown and Wolpert (1990)* as an important element of left-right patterning, in which asymmetric morphogenesis is local, and can be uncoupled from a global left-right biasing mechanism coordinating the position of different organs according to the same reference. This elegant model of left-right patterning is supported by the observation that in the absence (*Brennan et al., 2002*) or in case of bilateral expression (*Murray and Gridley, 2006*) of the left determinant Nodal, the process of asymmetric morphogenesis still takes place, that is, the heart does not remain symmetrical and some heart looping occurs, with a random orientation, probably due to stochastic and spontaneous left-right variations. Our computer simulations indeed show that the heart would only remain straight if there was a complete absence of asymmetry, a very unlikely situation in a noisy biological context. Simulations with our model (*Figure 6*) predict that very small localised differences in growth rate (in the order of 1%) will lead to significant buckling. The curvature is stronger with a progressive breakdown of a dorsal attachment than without (data not shown). The parameter space of left-right asymmetries compatible with the helix shape observed in vivo is narrow. A tight coordination between the intensities and positions of left-right asymmetry at either pole is required.

The looping mechanism that we propose for the heart tube shares similarities with that proposed for the looping of the embryonic gut (*Davis et al., 2008*). Left-right cellular asymmetries in the dorsal mesentery have been uncovered, with elegant cell-based computer simulations showing that this is sufficient to generate a leftward tilt of the gut. However, looping of the midgut is more than a tilting. Generation of a S-shape for the midgut has been proposed to depend on a buckling mechanism. Yet, the growth of the tube between fixed poles, and the opposite left-right deformation at the anterior pole have not been quantified. The similarities in the gut and heart raise the possibility of a common framework for the asymmetric morphogenesis of tubular organs. Our model at the tissue level will be generally useful to predict which tubular shape emerges from a given combination of mechanical constraints and left-right asymmetries. Together with the image analysis tools that we have developed to quantify the shape of the heart tube, it will now be possible to explore, in various experimental conditions, the parameter space in vivo and decipher the molecular and cellular determinants of asymmetric tubular morphogenesis.

## Materials and methods

### Key resources table

| Reagent type (species) or resource | Designation | Source or reference | Identifiers | Additional information |
|---|---|---|---|---|
| Strain, strain background (Mus musculus) | wild-type, Swiss background | Janvier | | |
| Strain, strain background (Mus musculus) | wild-type, C57Bl6 background | Janvier | | |
| Strain, strain background (Mus musculus) | T4nLacZ, Swiss background | *Biben et al. (1996)* doi:10.1006/dbio.1996.0017 | PMID 8575622 | |
| strain, strain background (Mus musculus) | Shh+/-, C57Bl6J background | *Gonzalez-Reyes et al. (2012)* doi:10.1016/j.neuron.2012.05.018 | MGI:5440762 | |
| Strain, strain background (Mus musculus) | Polr2aCreERT2/+, C57Bl6 background | *Guerra et al. (2003)* PMID:12957286 | MGI:3772332 | |

*Continued on next page*

*Continued*

| Reagent type (species) or resource | Designation | Source or reference | Identifiers | Additional information |
|---|---|---|---|---|
| Strain, strain background (Mus musculus) | R26YFP/+, C57Bl6 background | *Srinivas et al. (2001)* PMID:11299042 | MGI:2449038 | |
| Strain, strain background (Mus musculus) | R26Rtdtomato/+ (Ai14), C57Bl6 background | *Madisen et al. (2010)* doi:10.1038/nn.2467 | MGI:3809524 | |
| Antibody | anti-PH3 (rabbit monoclonal) | Abcam | Abcam: ab32107 | (1:100) |
| Antibody | anti-Isl1 (mouse monoclonal) | Developmental Studies Hybridoma Bank | DSHB: 39.4D5 | (1:50) |
| Antibody | anti-MMP2 (mouse monoclonal) | Santa Cruz | Santa Cuz: sc13594 | (1:50) |
| Antibody | goat anti-rabbit IgG Alexa Fluor 546 | Invitrogen | Invitrogen: A11035 | (1:500) |
| Antibody | goat anti-mouse IgG2b Alexa Fluor 488 | Invitrogen | Invitrogen: A21141 | (1:500) |
| Antibody | goat anti-mouse IgG1 Alexa Fluor 488 | Invitrogen | Invitrogen: A21121 | (1:500) |
| Commercial assay or kit | JB-4 embedding kit | Polysciences | Polysciences: 00226–1 | |
| Chemical compound, drug | GM6001 (Ilomast) | Millipore | Millipore: CC1000 | 10 µM |
| Software, algorithm | Gftbox | *Kennaway et al. (2011)* doi:10.1371/journal.pcbi.1002071 | | Matlab Finite Element Analysis package simulating biological growth |
| Software | ICY | de Chaumont et al. 2012 doi:10.1038/nmeth.2075 | | Open platform for bioimage informatics |

## Animal models

Control embryos (*Figure 1*) were from a mixed genetic background. The *Shh*$^{+/-}$ mouse line (*Gonzalez-Reyes et al., 2012*) was maintained in a C57Bl6J genetic background. *Shh*$^{+/+}$ and *Shh*$^{+/-}$ were indistinguishable and used together as control embryos. Animal procedures were approved by the ethical committee of the Institut Pasteur and the French Ministry of Research. For imaging, embryos were dissected, incubated in cold 250 mM KCl (at E9.5), fixed in 4% paraformaldehyde or Bouin's fluid, dehydrated and embedded in methacrylate resin, as previously described (*Weninger et al., 2006*). The number of somites was evaluated from the HREM images.

## HREM imaging

HREM acquires images of the surface of the resin block, in which the embryo is embedded, to produce perfectly registered digital image stacks capturing the 3D tissue architecture at high resolution. Resulting datasets comprise 1000–2000 images of 1 × 1 µm resolution produced by repeated removal of 1–2 µm sections. HREM was performed on E8.5 or E9.5 embryos as described previously (*Mohun and Weninger, 2012*), using the optical high-resolution episcopic microscope (Indigo Scientific).

## 3D reconstruction and quantification

Hearts were segmented from HREM images using the Imaris software (Bitplane). The contour of the myocardium was manually outlined at regular Z intervals, and the Create Surface function was used to reconstruct the 3D surface. The notochord was similarly segmented to serve as a reference longitudinal axis. 3D visualisation was produced with ICY (Institut Pasteur, Paris) for raw images, and with Blender (Blender Foundation, Netherlands) or DAZ Studio (Daz Productions Inc.) for segmented images. 3D PDF files were built with Acrobat Pro (Adobe Systems Inc.) after exporting the Imaris file in WRL format, and conversion into a U3D format with Meshlab (Visual Computing Lab).

The axis of the cardiac tube was reconstructed from the Imaris surface, using the Oblique Slicer function to intersect the tube perpendicularly, proceeding along its length. On each slice the polygon outlining the myocardium was drawn. The centroids of the successive polygons were computed with Matlab (geom3d library), and the 3D line exported in X3D format (figure2xhtml function). The X3D file was then imported in Blender and superimposed on the segmented myocardium. To quantify the average perimeter of the heart tube, 10 polygons, evenly distributed along the length, were selected and the perimeters of theoretical circular discs of the same area were taken as values. The tranverse sector, in which the tube axis is inscribed, was obtained after rotating the 3D line to superimpose the two extremities of the axis. The sector angle was measured from this view.

The orientation of the right ventricle–left ventricle axis was similarly obtained by intersection of the Imaris surface at the level of the interventricular sulcus. The polygon outlining the myocardium was drawn and its centroid computed. The axis was defined as the line perpendicular to this polygon. It was projected on the frontal plane, taken as perpendicular to the dorsal-ventral axis of the embryo. The orientation was measured as the angle between this projection and the notochord axis.

The thickness and the dorso-ventral elongation of the dorsal mesocardium, the positions of its ventral and dorsal aspects, as well as the angle between the heart tube and the dorsal pericardial wall, were measured in transverse sections, after aligning the HREM cubic images on the notochord axis with the ICY software (StackRotationByAngle plugin).

## Mouse embryo culture

E8.5 embryos from wild-type [Swiss] mice, or from the T4-nlacZ [Swiss] transgenic line (*Biben et al., 1996*) were collected, transferred to Hank's solution and labelled by injection of a lipophilic carbocyanine (Interchim, France) as described previously (*Domínguez et al., 2012*). Symmetrical dye injections were done at the right and left venous pole of the embryo using DiO and DiI to distinguish them. Injections at the most cranial midline of the heart tube were performed using DiI. Injected embryos were photographed using a Nikon Digital Sight DS-L1 camera system and a Nikon C-DSS230 stereomicroscope and, then, cultured for 24 hr in 75% rat serum, 25% T6 medium (*Whittingham, 1971*), with 5% $CO_2$, 5% $O_2$, 90% $N_2$ in rolling bottles in a precision incubator (BTC Engineering, Milton, Cambridge, UK). Two embryos were cultured per bottle, with one identified by an injection of DiI into the left headfold. At the end of the culture, embryos were washed in PBS, fixed 15 min in 4% paraformaldehyde in PBS, washed in PBS and kept at 4°C until examination with a Leica MZ16F fluorescence stereomicroscope. Embryos that showed widespread background heart fluorescence or appeared morphologically abnormal at the end of the culture, were excluded from the analysis.

For drug treatment, E8.5 embryos from wild-type [C57Bl6J] mice were collected. 10 µM of GM6001 (Ilomast - Millipore), or an equivalent volume of the adjuvant (DMSO), were added to the culture medium, in a 5% $CO_2$-5% $O_2$ atmosphere, and rinsed in PBS after 10 hr. Embryos were processed for HREM imaging or further incubated in culture medium, in a 5% $CO_2$-20% $O_2$ atmosphere, and harvested after 24 hr. Brightfield images were acquired with a Zeiss AxioCamICc5 Camera and a Zeiss StereoDiscovery V20 stereomicroscope.

## Immunofluorescence

Immunofluorescence on 10-µm cryostat sections was performed with a standard protocol, including permeabilisation in 0.75% Triton, blocking in 10% inactivated goat serum and 0.5% Triton, quenching of aldehydes in 2.6 mg/ml NH4Cl, and using primary antibodies to MMP2 (1/50, sc-13594), Alexa Fluor conjugated secondary antibodies (1/500) and Hoechst nuclear staining. Multi-channel 16-bit images were acquired with a Zeiss LSM 700 confocal microscope and 20X/0.75 or 40X/1.3 oil objectives.

Immunofluorescence on whole mount E8.5 embryos was performed using CUBIC clearing adapted from (*Susaki et al., 2015*). Samples were incubated overnight in the lipid-removing Reagent-1, then 48 hr with primary antibodies to PH3 (1/100, ab32107) and Isl1 (1/50, 39.4D5 DSHB), and 48 hr Alexa Fluor conjugated secondary antibodies (1/500, Molecular probes) and Hoechst nuclear staining in TSA Blocking Reagent (Perkin Elmer). Samples were finally incubated 48 hr in Reagent-2 for adjustment of the refractive index and mounted in 0.4% agarose in Reagent-2. Multi-channel 16-bit images were acquired with a Z.1 lightsheet microscope (Zeiss) and a 20X/1.0

objective. Automatic detection of mitotic cells was performed with the Spots plugin of Imaris and co-localisation with Isl1 staining was evaluated manually.

## Live-imaging

Live-imaging was performed as described by *Ivanovitch et al., 2017*. For labelling isolated cells, hydroxy-tamoxifen was administered by oral gavage (2–4 mg/ml) in *Polr2a$^{CreERT2/+}$* (*Guerra et al., 2003*); *R26R$^{tdtomato/YFP}$* ([*Srinivas et al., 2001*]; Ai14 line *[Madisen et al., 2010]*) mouse embryos at E7. Embryos were cultured under an upright LSM780 two-photon microscope equipped with a 5% $CO_2$ incubator and a 37°C heated chamber, in 50% fresh rat serum, 48% DMEM without phenol red, 1% N-2 neuronal growth supplement and 1% B-27 supplement, covered with mineral oil. Custom plastic holders were used to immobilise embryos during time-lapse acquisition, with the ventral side facing the objective. Multi-channel multi-section eight-bit images were acquired with a 20X/1 objective and MaiTai laser line at 1000 nm, every minute over 4 hr. The size of a scan was $512 \times 512 \times 19$ voxels, with a resolution of $0.83 \times 0.83 \times 4$ µm.

## FEA simulations

This model is based on the GFtbox software, a MATLAB (The Mathworks, Inc., USA) application developed for the simulation of a growing continuous sheet of tissue (*Kennaway et al., 2011*). The heart tube is represented as a cylindrical mesh, made of 938 nodes and 1800 finite elements, with two outside and inside surfaces and a thickness (*Figure 6*, *Figure 6—figure supplement 1A*). At each successive step during a simulation, each element is deformed according to a growth tensor field specified from the hypotheses of the model. The constraint of continuity of the tissue implies that the resulting growth is different from the input growth, this difference giving rise to residual strain. The output shape of the simulations is computed by minimising the energy derived from this residual strain, under the assumption of linear elasticity (see [*Kennaway et al., 2011*] for a detailed presentation and discussion of this modeling framework and its numerical implementation).

The simulations of a tube growing between fixed poles, in a minimal hypothesis (*Figure 2F–G*), were generated with the MATLAB code provided in *Source code 1*.

In a more refined model, the dorsal mesocardium was simulated as a displacement constraint on a set of nodes situated along two vertical lines on the dorsal side of the tube (*Figure 6*, *Figure 6—figure supplement 1B*). These nodes are not allowed any displacement along the y and z axes (the axes along which a thin beam would strongly resist deformation). This constraint is released to simulate breakdown of the dorsal mesocardium. The fixed distance between the two poles is implemented by forbidding any displacement along the z axis for the nodes situated at both extremities of the tube.

The local orientation of growth is defined relative to a reference axis of the tube, called the polarizer axis (*Figure 6—figure supplement 1C–D*), so that growth in any region may be defined by two components: longitudinal growth along the direction of the polarizer, and circumferential growth in a direction tangent to the surface of the tube and perpendicular to the direction of the polarizer. The direction of the polarizer is defined by the gradient of a morphogen diffusing from the venous pole towards the arterial pole, and thus represents the axis of the tube, which changes in direction and curvature over the course of the simulation.

The first 10 steps of the simulation are required to set up initial conditions, and establish the gradients that are used to define the polarizer as well as regions of the tube (right and left ventricles, venous, arterial, ventral, left regions), and thus smoothen the growth profile.

The six input parameters of growth were set as follows:

1. The basic longitudinal growth, active in every element of the tube, was set in order to obtain a 2.6-fold increase in the total length of the tube at the end of the simulation, as observed biologically between stages E8.5f and E8.5i (from 243 ± 62 µm to 638 ± 57 µm *Figure 2C*). The corresponding value is 2.5% per time step.
2. Until step 40, additional longitudinal growth was introduced on the ventral side of the tube at mid-length in order to reproduce the ventral bulge of the early heart tube. This leads to a peak value of 5% on the ventral spot, decreasing smoothly away from this spot (*Figure 6—figure supplement 1C*).
3. The initial rotation at the arterial pole is simulated by specifying, until step 30, a local circumferential growth that is positive on the left side and negative on the right side, so that the

arterial pole, being only constrained dorsally, effectively rotates clockwise. In line with biological observations (*Figure 4C*), the amount of circumferential growth is set at 1.1% per time step to produce a 25° rotation of the arterial pole, with a decreasing gradient towards mid-length of the tube (*Figure 6—figure supplement 1D*).

4. The asymmetric ingression of cells at the venous pole is simulated by specifying, between steps 30 and 60, a local longitudinal growth peaking at 7% per time step on the right side and decreasing smoothly to 2.5% on the left side (*Figure 6C*).

5. The inflation of the two ventricles is produced by an increased circumferential growth of 0.9% in the right ventricle and 1.4% in the left ventricle, with a smooth transition at the borders. In addition, the circumferential growth of the right ventricle is inhibited (resulting growth of 0.4%) in the dorsal left part of the tube (*Figure 6B*), to accentuate the inner curvature and produce a shape closer to biological observations. This is in agreement with proliferation gradients (*de Boer et al., 2012*) and the observations that cell proliferation is inhibited in compressed regions (see *Mammoto et al., 2013*).

Simulations were run for 90 steps. The MATLAB code containing the interaction function of the GFtbox model, and used to generate the shapes in *Figure 6*, is provided in *Source code 2*.

For exploring the parameter space of shapes shown in *Figure 7*, the same model was used, excluding only the inflation of the two ventricles. For *Figure 7A*, the arterial pole rotation was the same as above (25°) and the position of the burst at the venous pole varied between 0° and 360° (relative to dorsal) in increments of 90°. For *Figure 7B*, the position of the burst at the venous pole was fixed as in *Figure 6* (270° relative to dorsal). The arterial pole rotation varied between 0° and 60° by increasing the circumferential growth rate from 0% to 3% per time step. Asymmetry at the venous pole was increased in intensity by variations in the burst growth from 0% to 12%. Simulations were run for 80 steps. The MATLAB code used to generate the shapes is provided in *Source code 3*.

The simulation for *Figure 8A* replaced the rotation at the arterial pole by a burst of longitudinal growth, a similar mechanism to that of the venous pole asymmetry, positioned at 120° relative to the burst at the venous pole (peak of growth in a ventral left position). The MATLAB code used to generate the shapes is provided in *Source code 4*.

In simulations of a persistent dorsal mesocardium in *Figure 9*, the only change relative to the basic model was the absence of breakdown of the dorsal attachment, which thus was fixed in yz and free in x throughout the simulation. The MATLAB code used to generate the shapes is provided in *Source code 5*.

## Statistics

Sample size was checked post-hoc, using the PS software, in order to ensure a power of at least 0.8, with a type I error probability of 0.05, with an effect size of 100% (*Figures 2C*, *3F* and *4C*, *Figure 4—figure supplement 1B*), 50% (*Figure 5E*) or 25% (*Figures 5G* and *9F,G and K*). All sample numbers indicated in the text refer to biological replicates, i.e. different embryos. No outlier was excluded from the data analysis. Comparisons of two centre-values were done on the average, using a Student two-tailed test. A paired Student test was used for comparing left and right angles or lengths at successive positions (*Figure 4C*, *Figure 4—figure supplement 1B*). A Mann-Whitney test was used when a normal distribution could not be assumed. A chi-squared goodness-of-fit test, with Yates's correction for small sample size, was used to compare observed with expected distributions. An ANCOVA (analysis of covariance) was used to compare linear regressions. Tests were performed with either Excel or R statistical packages. When assessing whether a distance was significantly different from zero, confidence intervals were calculated assuming a normal distribution of measurements (*Figure 5B*). The experiments were not randomised and the investigators were not blinded to allocation during experiments and outcome assessment.

## Acknowledgements

We thank F Spitz for comments on the manuscript, AH Kottmann for the gift of the *Shh*$^{+/-}$ mouse line, N Gadessaud and the histology plateform of the SFR Necker, N Goudin and the imaging plateform of the SFR Necker, L Guillemot, J Xie, A Murukutla, J Eiblwieser, D Darby and C Cimper for technical assistance, C Ragni and H Moisset for preliminary observations, F Prin for advice on HREM and R Kennaway for adaptation of the modelling framework.

## Additional information

### Funding

| Funder | Grant reference number | Author |
| --- | --- | --- |
| Agence Nationale de la Recherche | 11-JSV2-00601 | Sigolène M Meilhac |
| Institut National de la Santé et de la Recherche Médicale | | Sigolène M Meilhac |
| Institut Pasteur | | Sigolène M Meilhac |
| Institut Imagine | | Sigolène M Meilhac |
| Spanish Ministry | BFU2015-71519-P | Miguel Torres |
| Human Frontier Science Program | LT000609/2015 | Kenzo D Ivanovitch |
| EMBO | ATL1275-2014 | Kenzo D Ivanovitch |

The funders had no role in study design, data collection and interpretation, or the decision to submit the work for publication.

### Author contributions

Jean-François Le Garrec, Conceptualization, Software, Formal analysis, Investigation, Visualization, Methodology, Writing—original draft, Project administration, Performed the quantitative analyses, Designed and performed computer simulations; Jorge N Domínguez, Investigation, Performed the DiI injection experiments; Audrey Desgrange, Investigation, Visualization, Methodology, Writing—review and editing, Generated HREM images, Performed dissections and the analysis of Mmp and cell proliferation; Kenzo D Ivanovitch, Investigation, Visualization, Performed live-imaging experiments; Etienne Raphaël, Formal analysis, Visualization, Contributed to the analysis of HREM images; J Andrew Bangham, Software, Designed the computer framework; Miguel Torres, Supervision, Funding acquisition, Writing—review and editing, Supervised live-imaging experiments; Enrico Coen, Software, Writing—review and editing, Designed the computer framework; Timothy J Mohun, Investigation, Generated HREM images; Sigolène M Meilhac, Conceptualization, Supervision, Funding acquisition, Investigation, Visualization, Methodology, Writing—original draft, Project administration

### Author ORCIDs

Jean-François Le Garrec https://orcid.org/0000-0002-2069-0251
Jorge N Domínguez https://orcid.org/0000-0003-4419-0929
Audrey Desgrange https://orcid.org/0000-0001-9716-755X
Kenzo D Ivanovitch https://orcid.org/0000-0003-4706-7686
Miguel Torres https://orcid.org/0000-0003-0906-4767
Enrico Coen https://orcid.org/0000-0001-8454-8767
Sigolène M Meilhac http://orcid.org/0000-0003-4080-2617

### Ethics

Animal experimentation: Animal procedures were approved by the Committee on the Ethics of Animal Experiments of the Institut Pasteur and the French Ministry of Research (Project APAFIS#6344-20 16021615059818 v5) and conducted under the supervision of an experimenter authorised by the Ministry of Agriculture (licence 75-1489 to S. Meilhac).

### Decision letter and Author response

Decision letter https://doi.org/10.7554/eLife.28951.031
Author response https://doi.org/10.7554/eLife.28951.032

## Additional files

**Supplementary files**

• Source code 1. Code used to generate *Figure 2F–G*.
DOI: https://doi.org/10.7554/eLife.28951.025

• Source code 2. Code used to generate *Figure 6* and *Video 2*.
DOI: https://doi.org/10.7554/eLife.28951.026

• Source code 3. Code used to generate *Figure 7*.
DOI: https://doi.org/10.7554/eLife.28951.027

• Source code 4. Code used to generate *Figure 8A*.
DOI: https://doi.org/10.7554/eLife.28951.028

• Source code 5. Code used to generate *Figure 9* and *Video 4*.
DOI: https://doi.org/10.7554/eLife.28951.029

• Transparent reporting form
DOI: https://doi.org/10.7554/eLife.28951.030

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
