## [Decision Letter]

Thank you for submitting your article "A predictive model of asymmetric morphogenesis from 3D reconstructions of mouse heart looping dynamics" for consideration by *eLife*. Your article has been favorably evaluated by Didier Stainier (Senior Editor) and three reviewers, one of whom is a member of our Board of Reviewing Editors. The reviewers have opted to remain anonymous.

The reviewers have discussed the reviews with one another and the Reviewing Editor has drafted this decision to help you prepare a revised submission.

Summary:

In this manuscript by Le Garrec et al., the authors study morphogenesis of the embryonic mouse heart. The authors have used high resolution episcopic microscopy (HREM) with 3D reconstruction to describe heart morphogenesis with a specific emphasis on the heart looping process. Using these 3D reconstructions the authors have quantified geometrical parameters of the developing heart. The authors have quantified where and when the mesocardium is broken down. Furthermore they describe when left-right asymmetries in the position of the heart tube could be observed. Based on these observations a mathematical model of the heart tube was reconstructed. In the model the observed arterial rotation, asymmetric cell addition at the venous pole and the breakdown of the mesocardium were implicated and their effects on heart morphogenesis were tested. From the modelling the authors conclude that all three aspects are required to form a helical heart tube. The prediction that the mesocardium break down is required for normal looping morphogenesis was validated in *shh* mutant mice, in which breakdown of the mesocardium was impaired as well as heart looping.

The topic of heart looping is of high interest because of the complexity of the process and the significance to congenital heart defects. The authors provide detailed descriptions of the process and use these to build a mathematical model that can be used to test hypothesis. This is an excellent, scholastic and well-written study that is likely to be a seminal paper in the field. The work contextualises disparate and sometimes perplexing findings such as randomisation of looping in the face of mutation of LR pathway genes, and the long-standing hypothesis of Brown and Wolpert 1990 that posits global and local determinants of organ LR morphogenesis.

However, there are some important issues to address.

Major Issues to be addressed:

1) Rotation of the arterial pole (related to Figure 4). The authors write:

'Heart looping is a directional event, which depends on left-right patterning. Thus, we examined early left-right asymmetries during the formation of the heart tube. The E8.5f stage, corresponding to the straight heart tube, had always been considered as a stage of bilateral symmetry, as seen externally (Figure 1). Strikingly, we found on HREM sections, that the heart tube at this stage was tilted towards the right side, so that it was not bilaterally symmetrical (Figure 4)'. This is probably not so striking given that asymmetric development of the anterior pole of the heart has been described previously in chick. Please put this into proper context.

2) The authors describe asymmetric tilting at the caudal aspect of the heart tube at stage E8.5g, which had been described earlier (PMID: 9192865). It is confusing that the authors use 'tilting' here and continue with 'rotation' in the next part or does one relate to posterior pole and the other anterior pole? Rotation of the heart tube has been described in chick (acknowledged by the authors), and fish (PMID: 18784369; PMID: 18267096), of which the latter are not mentioned. Appropriate credit should be given here.

3) In the graph of Figure 4 the rotation along the heart tube is shown. The rotation is positive for the arterial pole while being negative for the venous pole. The authors focus on the arterial pole rotation and ignore the negative rotation at the venous pole. This rotation observed at both poles would be consistent with the rotation/helix model proposed by Männer (PMID: 15103744). Please comment.

4) Related to results described in Figure 5: Results of the dye injections and their interpretation are confusing. a) The authors describe that they label both the right and left side of the mesoderm that will contribute to the heart tube. They score the embryos 24 hours after dye injection based on the distribution of the dyes in the heart tube. If only one of the labels sides contribute to the heart the result is scored as either right or left, and if no difference is observed they score these embryos as both left and right. Instead of showing the bar graph the authors should show the data in a table and include the three categories (instead of only showing two categories in the graph). It makes a difference whether the embryos labelled at E8.5g and scored as left were actually L>R or L=R. b) The observed asymmetric contribution from the right side was only observed in embryos labelled at E8.5g, at which stage the heart tube already shows asymmetric morphology. It seems very unlikely that this asymmetric contribution is the cause of the observed asymmetric morphology, as claimed by the authors, but rather is a consequence. c) In a previous study by Galli et al. (cited by the authors) a similar LR difference in cardiac contribution at the venous pole was described. It is unclear what the new data adds to this previous study

5) In Figure 8 the authors compare rotation of the arterial pole with asymmetric growth. The computer simulations for both these processes show a slight difference in the position of the ventral wall of the arteria pole. The authors use dye labelling of the ventral wall to look at this shift. The authors claim that they have observed a right-ward shift of the dye, but it is difficult to appreciate this shift in the static photographs. A more sensitive or dynamic technique, such as time-lapse imaging, should be used to address this. In addition, the authors have not addressed a possible mechanism that could cause this rotation, if it is not due to asymmetric growth.

6) The authors have used the *shh* mutant mouse model to validate the model's prediction about the role for the breakdown of the mesocardium. The question here is whether the *shh* mutant is the proper genetic model to do so. Besides the reduced breakdown of the mesocardium, *shh* mutant embryos have other phenotypes such as left-right patterning defects (PMID: 10500184)). In addition, *Shh* signalling is important for atrial septation due to its role in migration of cells from the second heart field (PMID: 19369393; PMID: 22898775). The authors try to address this by measuring tube length in the mutant and modelled heart as well as by modelling left-right differences. The latter is not convincing, because it is unclear why and how they use a 50% reduction in left-right patterning. In addition, the sum of all the effects is not tested in the model. Most convincing would be if the authors can (computationally?) rescue the mesocardium breakdown in the *shh* mutants and test if this would also rescue the heart looping defect.

---

## [Author Response]

Major Issues to be addressed:1) Rotation of the arterial pole (related to Figure 4). The authors write:'Heart looping is a directional event, which depends on left-right patterning. Thus, we examined early left-right asymmetries during the formation of the heart tube. The E8.5f stage, corresponding to the straight heart tube, had always been considered as a stage of bilateral symmetry, as seen externally (Figure 1). Strikingly, we found on HREM sections, that the heart tube at this stage was tilted towards the right side, so that it was not bilaterally symmetrical (Figure 4)'. This is probably not so striking given that asymmetric development of the anterior pole of the heart has been described previously in chick. Please put this into proper context.

We have removed the word “strikingly”, and repositioned our comment on the chick heart to introduce our results.

2) The authors describe asymmetric tilting at the caudal aspect of the heart tube at stage E8.5g, which had been described earlier (PMID: 9192865). It is confusing that the authors use 'tilting' here and continue with 'rotation' in the next part or does one relate to posterior pole and the other anterior pole? Rotation of the heart tube has been described in chick (acknowledged by the authors), and fish (PMID: 18784369; PMID: 18267096), of which the latter are not mentioned. Appropriate credit should be given here.

We have now clarified the wording. We use “tilting” (first section of the Results), in keeping with Biben et al., 1997, as a neutral description of the morphological sequence, referring to the orientation of the tube axis, rather than anything specific at a pole. We use the word “rotation” (fourth section of the Results), to provide a more mechanistic insight, where we have observed asymmetric angles of attachment of the tube, i.e. at the arterial pole.

We have now added a paragraph with appropriate references, to discuss potential homologies between the mouse and zebrafish models in jogging, rotation and looping of the heart (Discussion, third paragraph).

3) In the graph of Figure 4 the rotation along the heart tube is shown. The rotation is positive for the arterial pole while being negative for the venous pole. The authors focus on the arterial pole rotation and ignore the negative rotation at the venous pole. This rotation observed at both poles would be consistent with the rotation/helix model proposed by Manner (PMID: 15103744). Please comment.

We have clarified the text (subsection “Initial left-right asymmetries during mouse heart looping”, first paragraph). In the graph of Figure 4, statistical analyses indicate that the deviation at the arterial pole is statistically significant (max 25° rightward rotation, illustrated in Figure 4-value <0.001, paired Student test), whereas this is not the case at the venous pole (max 8° leftward rotation, illustrated in Figure 4>0.40 at E8.5f, p>0.10 at E8.5e). Therefore, our observations do not permit to conclude on a negative rotation of the venous pole. We have cited in 4 instances the alternative model, based on a negative rotation at the venous pole, proposed by Männer (2004) from simulations with rubber tubes. This previous model is similar to ours, for the principle of opposite deformations at the poles. However, our biological observations in the mouse would suggest that the underlying mechanisms are different at each pole.

4) Related to results described in Figure 5: Results of the dye injections and their interpretation are confusing. a) The authors describe that they label both the right and left side of the mesoderm that will contribute to the heart tube. They score the embryos 24 hours after dye injection based on the distribution of the dyes in the heart tube. If only one of the labels sides contribute to the heart the result is scored as either right or left, and if no difference is observed they score these embryos as both left and right. Instead of showing the bar graph the authors should show the data in a table and include the three categories (instead of only showing two categories in the graph). It makes a difference whether the embryos labelled at E8.5g and scored as left were actually L>R or L=R.

We now provide a table (Figure 5) showing the scoring of embryos in 3 categories (R>L, R=L and L>R). However, statistical analysis with a chi-square test consists in a comparison with a theoretical distribution. In the absence of any asymmetry, the expected proportions would be half R>L and half L>R (the null hypothesis). It is more accurate and more conservative to take into account the R=L cases, rather than discarding them. Thus, we decided to allocate cases of R=L according to the null hypothesis, as reported in the graph of Figure 5.

b) The observed asymmetric contribution from the right side was only observed in embryos labelled at E8.5g, at which stage the heart tube already shows asymmetric morphology. It seems very unlikely that this asymmetric contribution is the cause of the observed asymmetric morphology, as claimed by the authors, but rather is a consequence.

We have clarified the text (subsection “Initial left-right asymmetries during mouse heart looping”). At E8.5g, the tube is tilted towards the right, which reflects the rightward rotation of the arterial pole that we detect earlier at E8.5f. At E8.5g, the tube has not acquired a helical shape, which requires the additional leftward deformation at the venous pole. Thus, the asymmetric cell ingression that we observe at the venous pole at E8.5g is in keeping with the curvature of the atrioventricular region at E8.5h and later stages.

c) In a previous study by Galli et al. (cited by the authors) a similar LR difference in cardiac contribution at the venous pole was described. It is unclear what the new data adds to this previous study

We have clarified the text. Galli et al. (2008) identified LR asymmetry as a number of differentiated cells in explant studies of the posterior heart field at 6 somites. They also investigated cell proliferation asymmetry based on quantifications of PH3 staining in histological sections. They reported LR asymmetry in cell proliferation in the sinus venosus at 7 somites, but not in the Isl1 positive posterior heart field (between 5-7 somites). All embryos were staged according to somite numbers. Our work now addresses asymmetric cell ingression in vivo, based on symmetrical DiI labelling in 71 embryos. The embryos are staged with a novel nomenclature, better reflecting the progression of heart looping. It is essential for the computer model to reconstitute the kinetics of different parameters and shape changes and thus acquire data within the same staging nomenclature. For this reason, we have now re-investigated cell proliferation, with a higher spatio-temporal resolution, using our staging system and 3D images (Figure 5).

5) In Figure 8 the authors compare rotation of the arterial pole with asymmetric growth. The computer simulations for both these processes show a slight difference in the position of the ventral wall of the arteria pole. The authors use dye labelling of the ventral wall to look at this shift. The authors claim that they have observed a right-ward shift of the dye, but it is difficult to appreciate this shift in the static photographs. A more sensitive or dynamic technique, such as time-lapse imaging, should be used to address this. In addition, the authors have not addressed a possible mechanism that could cause this rotation, if it is not due to asymmetric growth.

We now provide novel data from time-lapse imaging of the heart tube in the mouse embryo, showing cell movement consistent with a rotation of the arterial pole (Figure 8—figure supplement 1 and Video 3). We have added hypotheses on the rotation mechanism in the Discussion (fourth paragraph).

6) The authors have used the shh mutant mouse model to validate the model's prediction about the role for the breakdown of the mesocardium. The question here is whether the shh mutant is the proper genetic model to do so. Besides the reduced breakdown of the mesocardium, shh mutant embryos have other phenotypes such as left-right patterning defects (PMID: 10500184)). In addition, Shh signalling is important for atrial septation due to its role in migration of cells from the second heart field (PMID: 19369393; PMID: 22898775). The authors try to address this by measuring tube length in the mutant and modelled heart as well as by modelling left-right differences. The latter is not convincing, because it is unclear why and how they use a 50% reduction in left-right patterning. In addition, the sum of all the effects is not tested in the model. Most convincing would be if the authors can (computationally?) rescue the mesocardium breakdown in the shh mutants and test if this would also rescue the heart looping defect.

We discuss the different roles of *Shh* in a specific paragraph of the Discussion. In the computer model, we used a 50% reduction in left-right patterning as it reproduces well the positioning of the RV relative to the LV. We have also tested in the computer model, the combination of 50% reduction in left-right patterning and persistent dorsal mesocardium, which does not lead to a simulated shape closer to that observed in *Shh^-/-^* mutants. The position of the right ventricle is more severely impaired by the double effect, and is thus not compatible with the situation in *Shh^-/-^* mutants.

Computationally, we have shown the shape produced with breakdown of the dorsal mesocardium (Figure 6), without any dorsal mesocardium (Figure 6—figure supplement 1) and with a persistent dorsal mesocardium (Figure 9). As an alternative to a rescue experiment, which does not make sense computationally, we provide a novel experimental condition, reproducing the persistence of the dorsal mesocardium without manipulating *Shh* (Figure 10). When matrix metalloproteases were inhibited by the drug GM6001, the dorsal mesocardium was persistent and looping was impaired in a similar way as predicted by the computer model.